# Activation of a transient progenitor state in the epicardium is required for zebrafish heart regeneration

Yu Xia[1,2,10], Sierra Duca[1,2,10], Björn Perder[1,2], Friederike Dündar [3,4], Paul Zumbo[3,4], Miaoyan Qiu[1,2], Jun Yao[1,2], Yingxi Cao[1,2], Michael R. M. Harrison [1,2], Lior Zangi[5,6,7], Doron Betel [4,8,9] & Jingli Cao [1,2] ✉

The epicardium, a mesothelial cell tissue that encompasses vertebrate hearts, supports heart regeneration after injury through paracrine effects and as a source of multipotent progenitors. However, the progenitor state in the adult epicardium has yet to be defined. Through single-cell RNA-sequencing of isolated epicardial cells from uninjured and regenerating adult zebrafish hearts, we define the epithelial and mesenchymal subsets of the epicardium. We further identify a transiently activated epicardial progenitor cell (aEPC) subpopulation marked by *ptx3a* and *col12a1b* expression. Upon cardiac injury, aEPCs emerge from the epithelial epicardium, migrate to enclose the wound, undergo epithelial-mesenchymal transition (EMT), and differentiate into mural cells and *pdgfra⁺hapln1a⁺* mesenchymal epicardial cells. These EMT and differentiation processes are regulated by the Tgfβ pathway. Conditional ablation of aEPCs blocks heart regeneration through reduced *nrg1* expression and mesenchymal cell number. Our findings identify a transient progenitor population of the adult epicardium that is indispensable for heart regeneration and highlight it as a potential target for enhancing cardiac repair.

Adult zebrafish possess a remarkable capacity for scarless heart regeneration after injury, which is achieved through the proliferation of existing cardiomyocytes (CMs)[1–4]. CM proliferation is aided by the cellular and molecular environment provided by non-muscle tissues, such as the epicardium, a mesothelial layer of vertebrate hearts[5–7]. The epicardium is a heterogeneous population containing stem cells or progenitors that convert into other supporting cell types, such as mural cells (i.e., smooth muscle cells and pericytes) and fibroblasts, during development and regeneration[5,8–11]. Following cardiac injury in

adult zebrafish, epicardial cells are activated to turn on embryonic genes, proliferate, and migrate to repopulate the wound site. In addition to supplying supporting cell types, epicardial cells provide paracrine signals and extracellular matrix (ECM) components for CM division and coronary angiogenesis[5–7,12]. Recent studies have highlighted the vital roles of the epicardium in zebrafish heart regeneration[13–15], and the mobilization of epicardial cells has been reported to improve mammalian heart repair[16,17]. However, the epicardial progenitor state during heart regeneration remains largely

[1]Cardiovascular Research Institute, Weill Cornell Medical College, 1300 York Avenue, New York, NY 10065, USA. [2]Department of Cell and Developmental Biology, Weill Cornell Medical College, 1300 York Avenue, New York, NY 10065, USA. [3]Department of Physiology and Biophysics, Weill Cornell Medical College, 1300 York Avenue, New York, NY 10065, USA. [4]Applied Bioinformatics Core, Weill Cornell Medical College, 1300 York Avenue, New York, NY 10065, USA. [5]Cardiovascular Research Institute, Icahn School of Medicine at Mount Sinai, New York, NY 10029, USA. [6]Department of Genetics and Genomic Sciences, Icahn School of Medicine at Mount Sinai, New York, NY 10029, USA. [7]Black Family Stem Cell Institute, Icahn School of Medicine at Mount Sinai, New York, NY 10029, USA. [8]Division of Hematology and Oncology, Department of Medicine, Weill Cornell Medical College, 1300 York Avenue, New York, NY 10065, USA. [9]Institute for Computational Biomedicine, Department of Medicine, Weill Cornell Medical College, 1300 York Avenue, New York, NY 10065, USA. [10]These authors contributed equally: Yu Xia, Sierra Duca. ✉e-mail: jic4001@med.cornell.edu

uncharacterized due to the lack of genetic tools to label and trace epicardial subsets.

In this work, for unbiased assessment of the epicardial progenitor state in regenerating hearts, we perform single-cell RNA-sequencing (scRNA-seq) of epicardial cells isolated from hearts undergoing regeneration. Using scRNA-seq analysis and genetic approaches, we define the epithelial and mesenchymal subpopulations and the mural lineage of the epicardium. We also identify a transiently activated epicardial progenitor cell (aEPC) population and define their molecular features. These aEPCs are indispensable for heart regeneration as they differentiate into mural cells and mesenchymal epicardial cells and supply pro-regenerative factors during regeneration. Therefore, our study reveals the active driver of epicardium-mediated heart regeneration and provides the basis for harnessing the epicardium for heart repair.

## Results

### Heterogeneity and EMT of epicardial cells in the adult heart

In zebrafish, *tcf21* is a widely used epicardial marker that labels both quiescent and active epicardial cells, while other epicardial markers, such as *tbx18* and *wt1*, only label part of the epicardium[18]. We first examined the *tcf21*[+] epicardial cell distribution in uninjured hearts, using a nuclear EGFP reporter (nucEGFP) driven by the regulatory sequence of *tcf21*[18]. As shown in Fig. 1a, b, *tcf21*[+] nuclei reside in multiple layers of the uninjured ventricular wall. While the outermost layer expresses *aldh1a2* (it is expressed in the endocardium as well; see also

ref. 18), the inner layers of *tcf21*[+] nuclei enter the compact muscle and are *aldh1a2*-negative (Fig. 1b, c). Thus, the *tcf21* reporters label 2 distinct epicardial subsets in the uninjured adult zebrafish heart: the epithelial epicardial cells (*aldh1a2*[+] outermost layer) and the mesenchymal epicardial cells that enter the compact muscle.

Previous studies have suggested that zebrafish epicardial cells undergo epithelial-mesenchymal transition (EMT) during heart regeneration[19,20]. However, EMT cannot be definitively identified without a transgenic tool specific to the epithelial layer of the epicardium. To address this, we developed an approach of pericardial sac injection of modified RNA (modRNAs)[21,22] for transient gene expression in the epicardium. We also created a bacterial artificial chromosome (BAC) transgenic line in which the *tcf21* regulatory sequences drive a Cre-releasable floxed BFP-stop cassette followed by an mCherry-NTR cassette (*tcf21:loxP-BFP-Stop-loxP-mCherry-NTR* or *tcf21:Switch* for short, Fig. 1d). By injecting Cre modRNAs into fish carrying the *tcf21:Switch* line, we labeled the epithelial layer of the epicardium with mCherry (Fig. 1d, e, uninjured). We did not observe a single labeled mesenchymal cell in the apex half of the uninjured ventricle 10 days after injection (Fig. 1f, Ctrl, 13 hearts analyzed). Although we could not rule out labels of deeper *tcf21*[+] cells upon injection, combined use of the *tcf21:Switch* line with the pericardiac cavity injection of Cre modRNA injection specifically limits our labeling to the epithelial epicardium at least by day 10 after injection in the adults. To monitor EMT of epicardial cells, Cre modRNAs were injected 3 days before the amputation injury, and hearts were collected at 7 days post

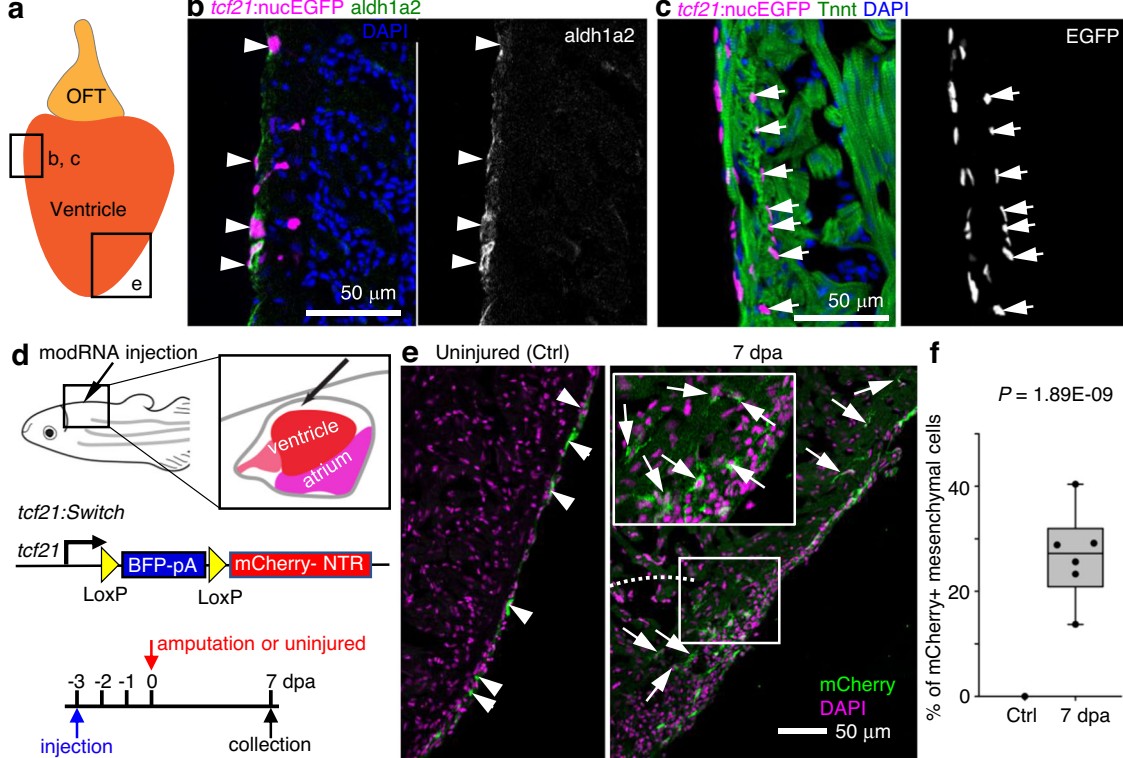

**Fig. 1 | Cellular heterogeneity and EMT of *tcf21*[+] cells in the zebrafish heart. a** Cartoon of an adult zebrafish heart showing the ventricle and outflow tract (OFT). The frames indicate representative regions for cryosection-section images in **b**, **c** and **e**. **b** A cryosection image of an uninjured heart showing *tcf21*:nucEGFP in magenta, antibody staining against aldh1a2 in green. Nuclei were stained with DAPI (blue). A single-channel image of aldh1a2 signals is shown in grayscale on the right. Arrowheads indicate EGFP[+]adh1a2[+] cells. Scale bar, 50 μm. **c** A cryosection image of an uninjured heart showing *tcf21*:nucEGFP in magenta, antibody staining against Tnnt in green. Nuclei were stained with DAPI (blue). A single-channel image of EGFP is shown in grayscale on the right. Arrows indicate EGFP[+] mesenchymal epicardial cells. Scale bar, 50 μm. **d** Schematic of experimental design for modRNA injection. **e** Section images of uninjured (left) and 7 dpa (right) hearts carrying the *tcf21:Switch* reporter at 10 days post Cre modRNA injection. Arrows and arrowheads indicate representative mCherry[+] mesenchymal and epithelial cells, respectively. Scale bar, 50 μm. **f** Quantification of mCherry[+] mesenchymal epicardial cells in the experiment of **e**. The largest cryosection of each heart was quantified for mCherry[+] cells in the apex half of the ventricle. n = 13 (Ctrl) and 6 (7 dpa), respectively. Two-tailed Student's t test. The box plot shows the median (center line), upper and lower quartiles (box limits), minimum and maximum values (whiskers), and individual values (points). Source data are provided as a Source data file.

amputation (dpa) to assess mCherry expression. We observed 26.8% on average of mCherry[+] cells entering the mesenchymal layer (Fig. 1e, f, 7 dpa), indicating an EMT process in which the epithelial epicardial cells give rise to the mesenchymal epicardial cells during heart regeneration.

## scRNA-seq reveals distinct epicardial subsets

To systemically dissect epicardial subsets, we applied scRNA-seq analysis of *tcf21*[+] cells during heart regeneration. Following cardiac injury in adult zebrafish, the epicardial cells turn on embryonic genes, a process called activation[5]. The epicardium is activated organ-widely at 3 dpa before restricting activation to the injury site starting at 7 dpa[19]. We injured *tcf21:nucEGFP* fish by partial ventricular amputation and collected ventricles at 3 or 7 dpa together with uninjured clutchmates (Ctrl). Single live nucEGFP[+] cells were isolated by Fluorescence-activated cell sorting (FACS) and subjected to library preparation and scRNA-seq on the 10x Genomics Chromium platform (Fig. 2a and Supplementary Fig. 1). Three samples were collected that comprised of 4970, 3743, and 6428 cells passing the quality control for Ctrl, 3 dpa, and 7 dpa, respectively (Fig. 2b). Unbiased clustering with all samples identified 12 clusters (Fig. 2c and Supplementary Data 1). Eight clusters are grouped as the core epicardial population (Fig. 2c, clusters 0–5, 7, and 9). Three clusters, including 8, 10, and 11, appear to represent contaminating non-epicardial cells (or doublets). Cells in cluster 8 express protein tyrosine phosphatase receptor type C (*ptprc*), *cd74a*, and macrophage expressed 1 (*mpeg1*), therefore representing immune cells[23–25]. Cluster 10 exhibits high expression levels of myosin light chain 7 (*myl7*) and troponin T type 2a (*Tnnt2a*), markers of CMs, indicating cardiac muscle identity[26,27]. Cluster 11 appears to comprise endocardial and endothelial cells enriched for Fli-1 proto-oncogene ETS transcription factor a (*fli1a*) and kinase insert domain receptor like (*kdrl*) expression (Fig. 2c, d, and Supplementary Fig. 2a, b)[28,29].

Besides these contaminating clusters, cluster 6 was identified as mural cells expressing platelet-derived growth factor receptor beta *pdgfrb* (Fig. 2c)[30,31]. Further analysis of the mural cells demonstrated enriched transcripts for known pericyte and vascular smooth muscle cell markers, including tropomyosin 1 (*tpm1*), notch receptor 3 (*notch3*), regulator of G protein signaling 5a (*rgs5a*), myosin heavy chain 11a (*myh11a*), actin alpha 2 smooth muscle (*acta2*), transgelin (*tagln*), ATP-binding cassette sub-family C member 9 (*abcc9*), and chemokine C-X-C motif ligand 12b (*cxcl12b*), as well as two recently identified mural cell markers NDUFA4 mitochondrial complex associated like 2a (*ndufa4l2a*) and potassium voltage-gated channel Isk-related family member 4 (*kcne4*)[32,33] (Supplementary Fig. 2c). While *ndufa4l2a* is restricted to a subset of mural cells, *kcne4* is highly expressed in mural cells and also present in the core epicardial clusters at lower levels. These gene expression profiles suggest diverse mural cell types (e.g., smooth muscle cells and pericytes) in our FACS isolated samples.

We next examined the temporal dynamics of the core epicardial clusters across all samples. Cluster 5 emerges at 3 dpa and is largely reduced by 7 dpa, while cluster 9 is mainly present at 7 dpa (Fig. 2e–g). Cluster 2 expanded during regeneration, while the percentages of the remaining clusters (other than 2, 5, and 9) decreased at 3 dpa, but rebounded by 7 dpa (Fig. 2e–g). Our gene expression analysis demonstrates that *tcf21* is an epicardial marker with broad expression across all core epicardial clusters and different states of injury (Fig. 2h and Supplementary Fig. 2e). Of note, we found a relatively lower *tcf21* expression level in the 3 dpa-specific cluster 5 than in other clusters (Fig. 2h, and Supplementary Fig. 2d, e). This seems to match our recent discovery that the Tcf21 binding motifs are enriched in chromatin regions with decreased accessibility in epicardial cells at 3 dpa, which may suggest a transition in cell state[34]. In contrast to the broad expression of *tcf21*, other known epicardial markers such as *tbx18*, *wt1b*, *sema3d*, and *aldh1a2* are only enriched in specific subpopulations[18,35]. For instance, *wt1b* expression is enriched in Cluster 1, whereas *tbx18* is relatively depleted in that cluster compared to others (Fig. 2h and Supplementary Fig. 2d). *sema3d* is mainly expressed in cluster 2, while *aldh1a2* is expressed in both clusters 2 and 5. In addition, vascular endothelial growth factor Aa (*vegfaa*), a pro-angiogenic factor that was reported to be expressed by the epicardium and endocardium upon heart injury[36], is enriched in the mural cells and core epicardial clusters 0 and 1 (Fig. 2h and Supplementary Fig. 2a). Collagen type I alpha 2 (*col1a2*), a marker of the epicardium and cardiac fibroblasts[37], is highly expressed in all clusters, including the mural cells (Fig. 2h). In all, these results suggest a dynamic cellular heterogeneity of the epicardium and its derivatives during regeneration, and that clusters 5 (predominantly 3 dpa cells) and 9 (7 dpa) are likely the injury-induced pro-regenerative subsets.

## The epithelial and mesenchymal epicardium

To explore the heterogeneity within the core epicardial populations, we identified marker genes that distinguish these clusters. Clusters 0 and 1 have enriched expression of the pro-angiogenic factor *vegfaa* (Fig. 2h and Supplementary Fig. 2a)[36], and clusterin (*clu*) is highly expressed in clusters 3, 7, and 9 (Fig. 2d). *sema3d*, *aldh1a2*, and podocalyxin-like (*podxl*) define cluster 2 (Figs. 2h and 3a). Podxl was reported to localize to the apical plasma membrane of epithelial or endothelial cells[38,39], and thus is a sign of epithelial identity in the epicardium. The hyaluronic acid-organizing factors hyaluronan and proteoglycan link protein 1a (*hapln1a*), the cardiac mesenchymal stem cell and cardiac fibroblast marker platelet-derived growth factor receptor alpha (*pdgfra*), as well as the myocardial mitogen neuregulin 1 (*nrg1*) mainly label clusters other than 2 and 5 (Figs. 2h and 3a)[13,40,41]. Interestingly, *podxl* and *aldh1a2* label clusters (clusters 2 and 5) distinct from the *hapln1a*[+]*pdgfra*[+] population (0, 1, 3, 4, 7, and 9; Fig. 3a, b). HCR staining results indicate that *podxl* is expressed in the epithelial layer of the epicardium (Fig. 3c). By contrast, *hapln1a* is expressed by the inner layer of *tcf21*[+] epicardial cells, which represent the mesenchymal layer (Fig. 3d). In agreement with this finding, a recent study showed that *hapln1a* is expressed in an epicardial subset residing in the compact muscle that mediates hyaluronic acid (HA) secretion and myocardial regeneration[13]. Thus, the *hapln1a*[+]*pdgfra*[+] cells (clusters 0, 1, 3, 4, 7, and 9) are mesenchymal epicardial cells. Moreover, we noticed that a subset of the 3 dpa-specific cluster 5 expresses *podxl* (Fig. 3a), while the remainder of cluster 5 cells instead express relatively high levels of *snai1a* (Fig. 3e), likely representing cells undergoing EMT. A re-clustering and focused analysis on only the core epicardial clusters demonstrated the same findings (Supplementary Fig. 3 and Supplementary Data 2). These analyses suggest that cluster 5 is a transitional population, and that a subset of these cells is likely undergoing an EMT process.

### *ptx3a* and *col12a1b* label a transient epicardial subtype

We next focused on the markers of the 3 dpa-specific cluster 5 to characterize its identity. This cluster is enriched with pro-regenerative ECM and related genes including pentraxin 3 long a (*ptx3a*), collagen type XII alpha 1b (*col12a1b*), myristoylated alanine-rich protein kinase C substrate b (*marcksb*), and fibronectin 1a (*fn1a*) (Fig. 3e and Supplementary Fig. 4)[42–48]. In addition to the injury-induced epicardial expression of *fn1a* that is required for heart regeneration[42], collagen XII (Col XII) deposition was reported to be boosted in both the epicardium and wounded tissues after cryoinjury in zebrafish[43]. However, the cellular sources of Fn and Col XII within the epicardial population were unclear. Col XII is also an axon growth-promoting ECM that helps zebrafish spinal cord regeneration[44,45]. Ptx3 is a secreted humoral innate immunity factor that orchestrates inflammation and tissue repair[46]. Besides the interactions with pathogens and complement molecules, Ptx3 also interacts with ECM components, such as fibrin and plasminogen, to promote a timely removal of fibrotic ECM for

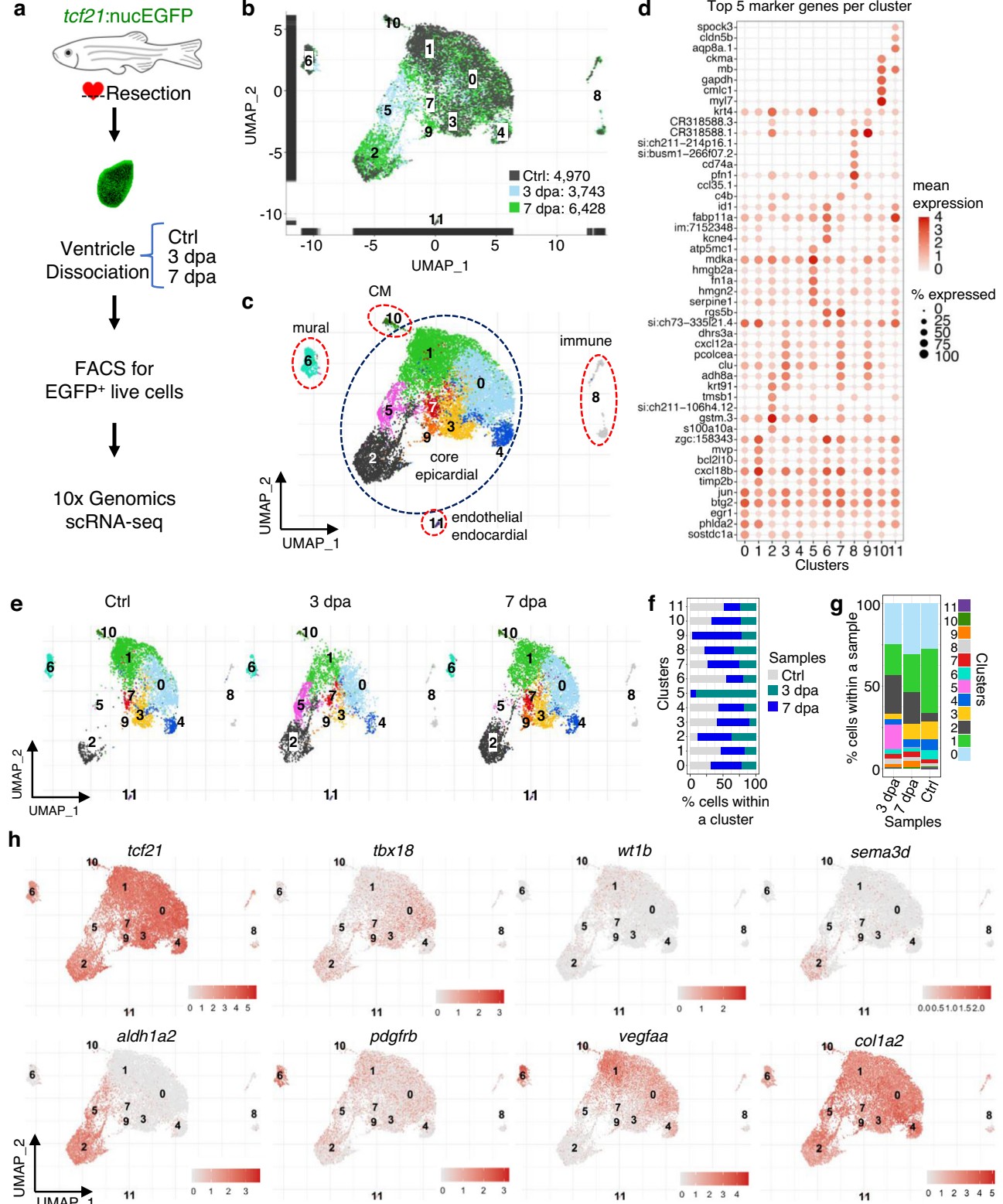

**Fig. 2 | ScRNA-seq reveals distinct subsets of the ventricular epicardial lineage during heart regeneration. a** Experimental design for single epicardial cell isolation and transcriptome sequencing. **b** UMAP of three samples combined. After the removal of droplets with very few genes as well as genes that could not be detected in at least 5 cells, the final dataset comprised 4970, 3743, and 6428 cells for Ctrl, 3 dpa, and 7 dpa, respectively. **c** UMAP of cell clusters with inferred cellular identities. **d** Expression of top 5 cluster marker genes across different clusters. This dot plot depicts the abundance and expression magnitude of individual genes across cells of given clusters. The size of the dot represents the fraction of cells with at least one UMI of the specified gene. **e** UMAPs showing changes of clusters across samples. **f** Proportions of the samples per cluster. **g** Cluster proportions per sample. **h** Normalized expression of top marker genes on UMAPs (3 samples combined).

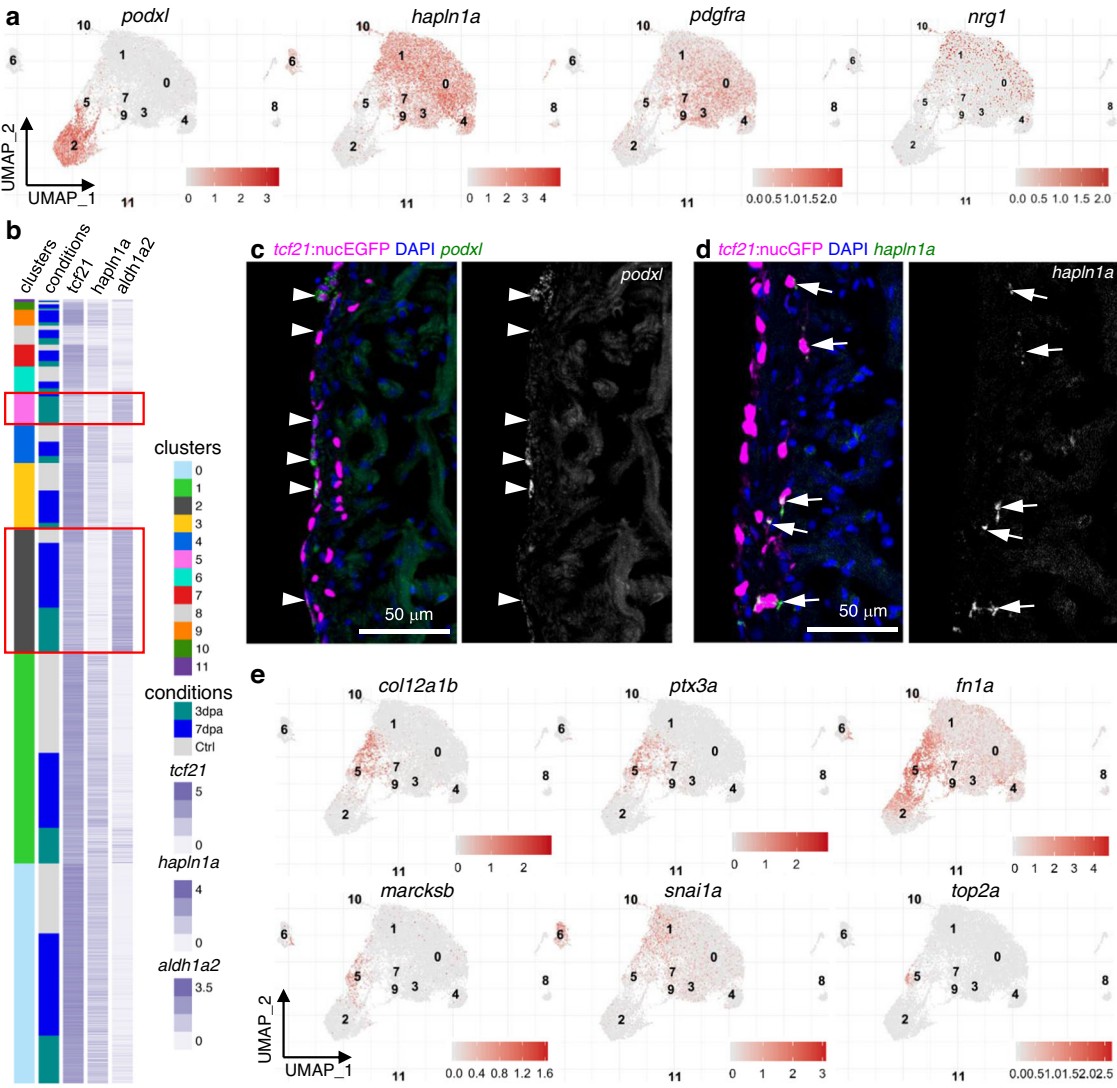

**Fig. 3 | The epithelial and mesenchymal subsets of the epicardium.**
**a** Normalized expression of marker genes on UMAPs (3 samples combined).
**b** Heatmap of 3 marker genes: *tcf21*, *aldh1a2*, and *hapln1a*. Normalized expression values are shown; cells were sorted by cluster membership. Clusters 2 and 5 are highlighted in red frames. **c** Images of a 3 dpa heart section showing the ventricular wall with *tcf21*:nucEGFP in magenta and HCR staining of *podxl* in green. Nuclei were stained with DAPI (blue). A single-channel image of *podxl* is shown in grayscale on

the right. Arrows indicate representative EGFP⁺*podxl*⁺ cells. Scale bar, 50 μm.
**d** Images of a 3 dpa heart section showing the ventricular wall with *tcf21*:nucEGFP in magenta and HCR staining of *hapln1a* in green. Nuclei were stained with DAPI (blue). A single-channel image of *hapln1a* is shown in grayscale on the right. Arrows indicate representative EGFP⁺*hapln1a*⁺ cells. Scale bar, 50 μm. **e** Normalized expression levels of marker genes for the 3 dpa-specific cluster on UMAPs (3 samples combined).

efficient tissue repair[49,50]. Additionally, Ptx3 was reported to have a cardioprotective function after acute myocardial infarction[51]. MARCKS is a ubiquitous substrate for protein kinase C and regulates the secretion of different substances. It has been shown to be highly upregulated during optic nerve regeneration in zebrafish, lens regeneration in newts, and cardiac tissue regeneration following infarction in mice[47,48,52]. Other cluster 5 enriched top markers include high mobility group box 2b (*hmgb2b*), ATP synthase membrane subunit c locus 1 (*atp5mc1*), serpin peptidase inhibitor clade H member 1a (*serpinh1a*), proteasome 20S subunit beta 1 (*psmb1*), and heat shock protein 90 beta member 1 (*hsp90b1*) (Supplementary Fig. 5). In addition, almost all proliferating cells (*top2a*⁺, a G2/M phase marker) are within the 3 dpa-specific cluster 5 (Fig. 3e), suggesting that cluster 5 is likely the primary cellular driver to restore the epicardial population after amputation injury. For an overview of the biological functions of marker genes for each cluster, we performed Gene Ontology (GO) enrichment analysis with specifically enriched transcripts in each core epicardial cluster (Supplementary Fig. 3e and Supplementary Data 3).

Notable enriched GO terms in the 3 dpa-specific subset include ECM organization, regeneration, metabolic processes, and translation. These results suggest that the 3 dpa-specific cluster is a pro-regenerative subset that mediates ECM remodeling, immune responses, and epicardial cell repopulation.

We next characterized marker gene expression for the 3 dpa-specific cluster through HCR staining. *ptx3a* is undetectable in the epicardium in uninjured hearts (Fig. 4a). Upon amputation injury, *ptx3a* is initially expressed in the *tcf21*⁺ epithelial layer of the entire ventricular epicardium at 1 dpa (Fig. 4b). By 3 dpa, the epicardial *ptx3a* transcripts are mostly restricted to the injury site in both the epithelial and mesenchymal layers of the epicardium, labeling the leading front of the regenerating *tcf21*⁺ cells in the wound region (Fig. 4c and Supplementary Fig. 4). Interestingly, *tcf21*:nucEGFP expression is reduced in these *tcf21*⁺*ptx3a*⁺ leader cells (Fig. 4c), which recapitulates the scRNA-seq result that the 3 dpa-specific cells have relatively lower *tcf21* expression compared to cells of the other core clusters (Fig. 2h). This suggests changes in cell state including cell proliferation. At 7 dpa, the

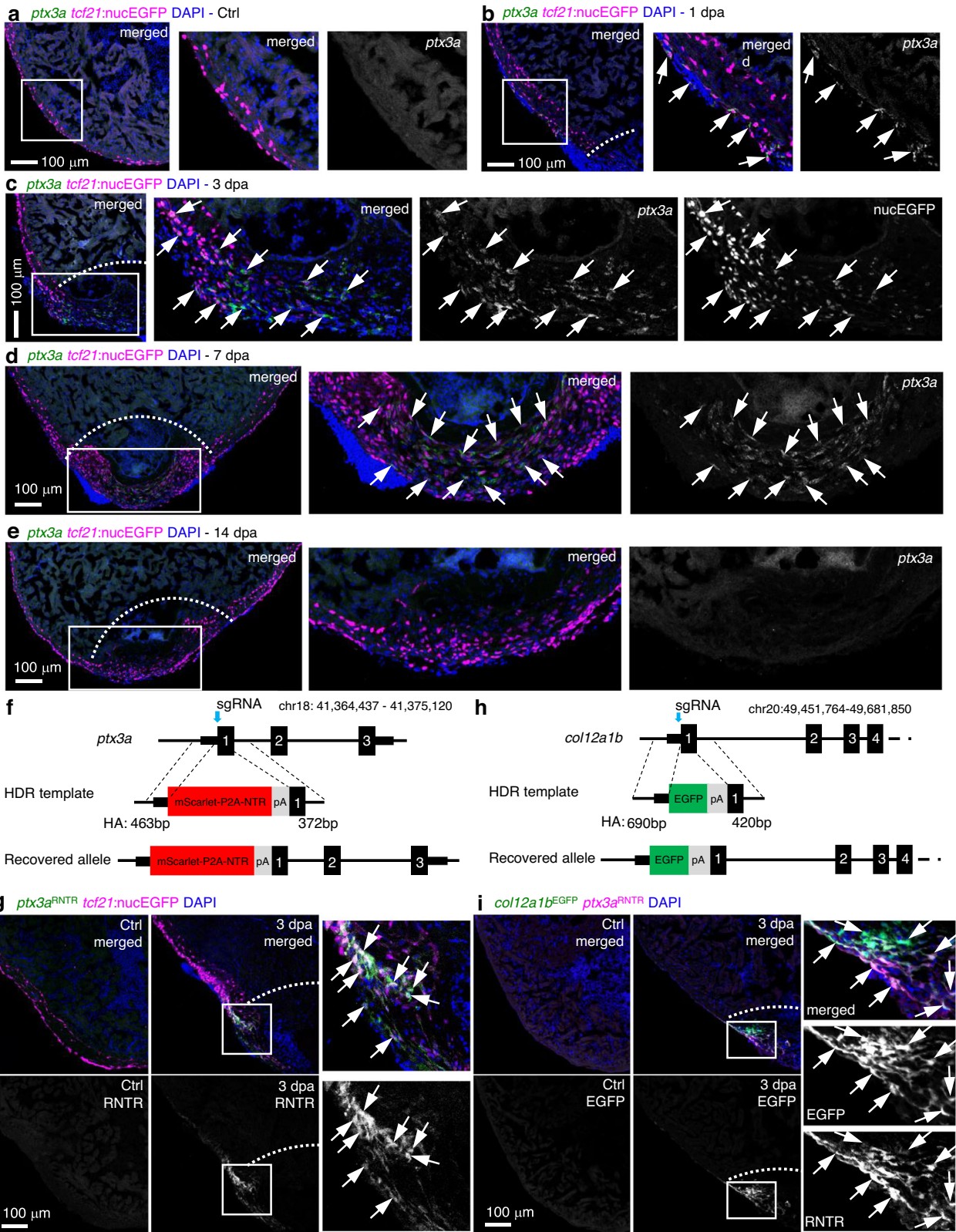

injury site is repopulated with predominantly *tcf21⁺* cells, and *tcf21:nucEGFP* expression is comparable to the flanking regions. However, only the cells in the most newly regenerated region of the apex are *ptx3a⁺* (Fig. 4d), further suggesting that *ptx3a* marks the leading front of the regenerating epicardium. The expression levels of *ptx3a* peak at 3 dpa, decrease at 7 dpa, and are minimal by 14 dpa (Fig. 4e). The results of *col12a1b* HCR staining demonstrated an expression

pattern similar to *ptx3a*, as suggested by our scRNA-seq analysis (Supplementary Figs. 4 and 6). These results indicate that the 3 dpa-specific cluster labeled by *ptx3a* and *col12a1b* represents a transient and pro-regenerative epicardial cell population.

To aid subset labeling and tracing, we generated knock-in alleles using the Crispr/Cas9 technique. A mScarlet-P2A-NTR-polyA cassette was inserted right after the start codon of the *ptx3a* gene using a

**Fig. 4 | *ptx3a* and *col12a1b* label a transient pro-regenerative epicardial sub-type. a–e** HCR staining results of *ptx3a* (green) on heart sections collected at 1 (**b**), 3 (**c**), 7 (**d**), and 14 dpa (**e**) together with the uninjured control (Ctrl, **a**). *tcf21*:nucEGFP (magenta) labels the epicardial cells. Nuclei were stained with DAPI (blue). Single-channel images show signals of *ptx3a* or nucEGFP. White dashed lines indicate the injury sites. The framed regions are enlarged to show details on the right of each panel. Arrows denote representative *ptx3a*⁺EGFP⁺ cells. Scale bars, 100 μm. **f** Schematic for generating the knock-in alleles for *ptx3a*. The gRNA binding site is marked with a cyan arrow. **g** The *ptx3a*^RNTR reporter recapitulates expression of *ptx3a* in the injured heart (3 dpa). No conclusive epicardial expression was observed in the uninjured heart (Ctrl). *tcf21*:nucEGFP (magenta) labels the epi-cardial cells. Nuclei were stained with DAPI (blue). Single-channel images show

signals of *ptx3a*^RNTR (with anti-DsRed antibody staining) at the bottom. White dashed lines indicate the injury sites. The framed regions are enlarged to show details on the right. Arrows denote representative RNTR⁺EGFP⁺ cells. Images of additional timepoints are in Supplementary Fig. 7. Scale bar, 100 μm. **h** Schematic for generating the knock-in alleles for *col12a1b*. The gRNA binding site is marked with a cyan arrow. **i** Section images showing *col12a1b*^EGFP reporter expression in green and *ptx3a*^RNTR expression in magenta (with anti-DsRed antibody staining). No epicardial EGFP expression was observed in the uninjured heart (left). Nuclei were stained with DAPI (blue). Single-channel images show signals of *col12a1b*^EGFP or *ptx3a*^RNTR. White dashed lines indicate the injury sites. The framed regions are enlarged to show details on the right. Arrows denote representative EGFP⁺mCherry⁺ cells. Scale bar, 100 μm.

double-stranded HDR template that has 463 bp and 372 bp homology arms flanking the cassette. A seamless insertion allele *ptx3a*^mScarlet-P2A-NTR (*ptx3a*^RNTR for short) was recovered through genotyping and sequencing (Fig. 4f). mScarlet signals in *ptx3a*^RNTR hearts emerge in *tcf21*⁺ cells from 1 dpa, peak at 3 dpa in the regenerating epicardial cells flanking the injury site, decrease at 7 dpa, and are barely detectable by 14 dpa (Fig. 4g and Supplementary Fig. 7), recapitulating the endogenous *ptx3a* expression pattern. Of note, the lower mScarlet signals at 1 and 2 dpa is only visible after an anti-DsRed antibody staining. A *col12a1b*^EGFP allele was also generated using the same strategy to visualize the endogenous *col12a1b* expression (Fig. 4h, i, and Supplementary Fig. 8). We demonstrate that both *ptx3a*^RNTR and *col12a1b*^GFP drive the same expression pattern as the genes *ptx3a* and *col12a1b*: labeling the 3 dpa-specific cluster after heart injury with no apparent epicardial expression in the uninjured adult hearts. Thus, we can define the transient pro-regenerative epicardial subtype labeled by expression of *ptx3a* or *col12a1b*.

### *ptx3a*⁺*col12a1b*⁺ aEPCs differentiate into mural cells

To infer the origin and fates of the *ptx3a*⁺*col12a1b*⁺ subset, we applied the Monocle3 trajectory reconstruction algorithm to all clusters[53]. We found that the 3 dpa-specific subset sits on a branching point leading to the epithelial cluster 2 (branch a), part of the mural population (cluster 6, branch b), and the mesenchymal subsets (branch c, Fig. 5a, b). This suggests that the 3 dpa-specific cluster is a progenitor state with the potential to give rise to different types of cells, which we thus named the activated epicardial progenitor cell (aEPC) population. To define the origin of these aEPCs, zebrafish carrying the *tcf21*:Switch;*tcf21*:CreER^t2;*col12a1b*^EGFP reporters were treated with 4-Hydroxytamoxifen (4HT) at both the embryonic (1 to 5 days post fertilization (dpf)) and adult stages (from 6 to 4 days before the heart injury, Fig. 5c). Hearts were collected at 3 dpa to assess colocalization of the mCherry and GFP signals. As shown in Fig. 5d, all EGFP⁺ cells around the wound are mCherry⁺, suggesting that these *col12a1b*⁺ aEPCs are derived from the spared epicardial cells upon injury, particularly from the epithelial layer of *tcf21*⁺ cells (i.e., cluster 2, *podxl*⁺).

Interestingly, we noticed a standalone population of mural cells (branch "d") in addition to the aEPC-derived branch "b" (Fig. 5b). A recent study found that *pdgfrb*⁺ cardiac mural cells are originated from the epicardium during heart development[30]. However, whether epicardial cells give rise to *pdgfrb*⁺ cells during heart regeneration is still unclear. Gene expression analysis indicates that the branch "b" mural cells express *fn1a*, while the branch "d" cells are mostly negative (Fig. 5e). Because *fn1a* expression is restricted to the injury site after 1 dpa[42], we hypothesized that the branch "b" (*fn1a*⁺) mural cells in the wound are derived from aEPCs. We next crossed the *tcf21*:H2A-mCherry (or *tcf21*:H2R for short) line with a *pdgfrb*:EGFP reporter[30]. Upon heart injury, we observed *pdgfrb*:EGFP⁺*tcf21*:H2R⁺ cells in the wound at 7 dpa (Fig. 5f), further suggesting an epicardial origin of these mural cells in the wound. To confirm the differentiation capacity of aEPCs to mural cells, we generated a *ptx3a*:CreER^t2 BAC line and crossed it with the *ubi*:loxP-EGFP-loxP-mCherry (*ubi*:Switch) line[54]. Adult zebrafish carrying

the *ubi*:Switch;*ptx3a*:CreER^t2 reporters were treated with 4HT from 2 to 5 dpa, and hearts were collected at 14 dpa for whole-mount HCR staining of *pdgfrb* (Fig. 5g). We observed mCherry⁺*pdgfrb*⁺ cells in the injury site, confirming an aEPC-to-mural differentiation (Fig. 5h). These results support the scRNA-seq-inferred notion that the 3 dpa-specific cluster entails activated epicardial progenitor cells that can give rise to mural cells in the regenerated hearts.

### aEPCs give rise to the epithelial and mesenchymal epicardium

To further characterize the additional aEPC differentiation potential for the mesenchymal epicardium (branch "c" in Fig. 5b), we focused on cells of the core clusters for which we observed dynamic gene expression patterns along the pseudotime branches "a" and "c" (Supplementary Fig. 3g). The epithelial epicardium initially expresses *fn1a*, *ptx3a*, and *col12a1b* to become aEPCs, followed by a transition to *hapln1a*-expressing mesenchymal cells. To confirm these gene expression dynamics, we performed HCR staining of *hapln1a* and *ptx3a*. As shown in Fig. 6a, cells expressing *hapln1a* in the mesenchymal epicardium lag behind the *ptx3a*⁺ leader cells that repopulate the wound at 3 dpa. At 7 dpa, *hapln1a* expression is enriched in the regenerated epicardial cells that flank the *ptx3a*⁺ cells in the wound (Fig. 6b). By 14 dpa, the regenerated *tcf21*⁺ cells in the wound are *hapln1a*⁺ but *ptx3a*⁻. These observations support the notion that aEPCs are the first responders in regenerating the epi-cardium, and that *hapln1a* likely marks the mature mesenchymal epicardium. To confirm the differentiation of aEPCs to *hapln1a*⁺ cells and the final fate of aEPCs upon completion of regeneration, adult zebrafish carrying the *ubi*:Switch;*ptx3a*:CreER^t2 reporters were treated with 4HT from 2 to 5 dpa, and hearts were collected at 30 dpa for whole-mount HCR staining of *hapln1a* and *podxl* (Fig. 6d). We observed prominent colocalization of mCherry with *hapln1a* or *podxl* expression at 30 dpa (Fig. 6e–g). The mCherry⁺ cells reside in both the epithelial (Fig. 6e–g, arrowheads) and mesenchymal layer (Fig. 6e–g, arrows) of the regenerated epicardium, indicating a final fate of aEPCs to both epicardial layers (Fig. 6h). In conclusion, our cell tracing experiments demonstrate that upon injury, aEPCs are derived from the epithelial epicardium and serve as the cellular source to regenerate all three epicardium-derived subsets, including mural cells and epithelial and mesenchymal epicardial cells.

### aEPCs are indispensable for heart regeneration

To test the requirement of aEPCs for heart regeneration, we used the *ptx3a*^RNTR line to ablate aEPCs. Since the aEPCs emerge at 1 dpa and their number peaks at 3 dpa, we applied amputation injury and then bathed fish in 5 mM metronidazole for 3 successive days from 3 to 5 dpa with fish water being changed daily[15]. Hearts were collected at 7 dpa for analyses (Fig. 7a). Blood clots and large pieces of extra tissues were observed in the aEPC ablated hearts (13 of 14 hearts analyzed), while there is only a minor noticeable wound in the vehicle-treated NTR-positive group (12 of 18 hearts analyzed) and the Mtz-treated NTR-negative group (11 of 16 hearts analyzed, Fig. 7b). Remarkably, *tcf21*⁺ cells failed to repopulate the wound region after aEPC ablation,

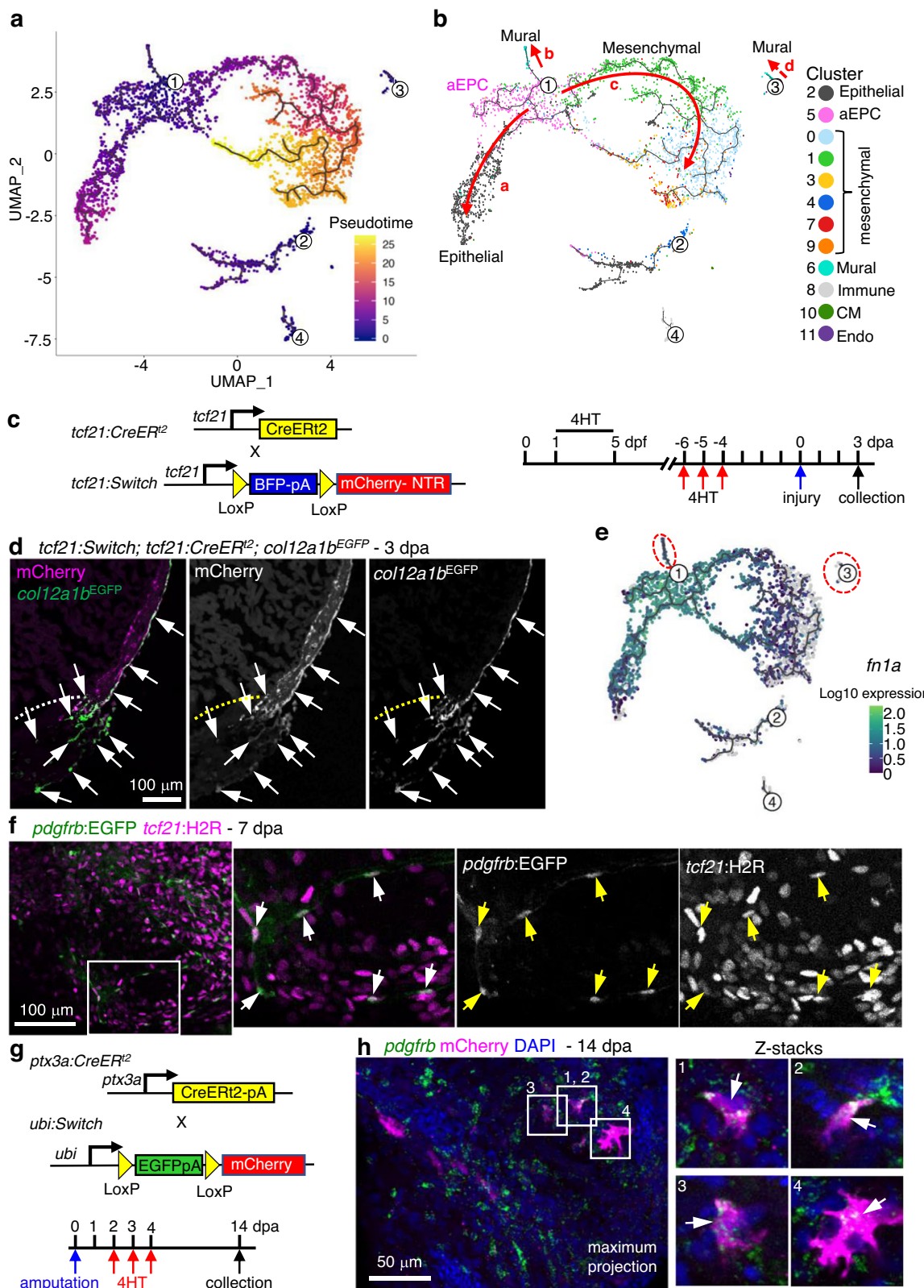

indicating impaired wound closure (Fig. 7c, d). CM proliferation, a hallmark of heart regeneration, is largely suppressed after aEPC ablation (~54% reduction compared to both control groups; Fig. 7e, f). By 30 dpa, Acid Fushin Orange G (AFOG) staining results indicate resolved fibrin and collagen deposition in the wound of vehicle-treated and Mtz-treated NTR-negative hearts. In contrast, prominent scar tissue is observed in 10 of 13 aEPC-ablated hearts, indicating failed regeneration

(Fig. 7g, h). These results suggest that aEPCs are indispensable for successful heart regeneration.

## aEPCs are the source of epicardial progenies and factors

Recent single-cell analyses of epicardium and epicardial-derived cells have identified pro-regenerative subpopulations in the adult heart. To assess how aEPCs relate to these subpopulations, we re-analyzed the

**Fig. 5 | aEPCs give rise to *pdgfrb*⁺ mural cells during regeneration. a, b** Cell trajectories suggested by pseudotime analysis with Monocle 3. Shown on a UMAP, the starting point of each trajectory was labeled with a number. The pseudotime is shown as a heatmap in **a**. Red arrows and letters highlight different branches in **b**. Clusters are labeled in the same number and color as in Fig. 2c. **c** Schematic of transgenic lines and experimental design to define the origin of aEPCs. 4HT was used at 10 µM. **d** Section images of 3 dpa hearts carrying the *tcf21:Switch; tcf21:CreER^t2; col12a1b^EGFP* reporters. 4HT treatment was performed as indicated in **c**. mCherry and EGFP are shown in magenta and green, respectively in the merged image. Single-channel images are shown in grayscale. Arrows indicate representative EGFP⁺mCherry⁺ cells. White dashed lines indicate the injury sites. Scale bar, 100 µm. **e** *fn1a* expression shown on the pseudotime UMAP. The mural cell trajectories are circled with red dashed lines. **f** Whole-mount images of the ventricular surface showing expression of *pdgfrb:EGFP* (green) and *tcf21:H2R* (magenta) in transgenic lines at 7 dpa. The framed regions are enlarged to show details on the right, with single-channel images shown in grayscale. Arrows indicate representative EGFP⁺mCherry⁺ cells. Scale bar, 100 µm. **g** Schematic of transgenic lines and experimental design to define the fate of aEPCs. 4HT was used at 5 µM. **h** Whole-mount images of the ventricular surface from hearts carrying the *ubi:Switch; ptx3a:CreER^t2* reporters. *pdgfrb* expression is detected by HCR staining (green). mCherry is shown in magenta, and nuclei were stained with DAPI (blue). Arrows indicate representative *pdgfrb*⁺mCherry⁺ cells. A maximum projection image is shown on the left. Z-stack images of the numbered frames are shown on the right. Scale bar, 50 µm.

published scRNA-seq datasets and the defined pro-regenerative genes. Kapuria et al. performed scRNA-seq of FACS-isolated *pdgfrb*⁺ mural cells from injured adult hearts and found that epicardium-derived *pdgfrb*⁺ mural cells are essential for coronary development and heart regeneration (Supplementary Fig. 9a, b)[30]. We demonstrated in our current study that at least part of these *pdgfrb*⁺ mural cells in the injury site are derived from aEPCs in adult hearts (Fig. 5). Sun et al. recently profiled single *tcf21*⁺ cells isolated from adult hearts upon CM ablation[13]. They found that an *hapln1a*⁺ subset providing hyaluronic acid (HA) is required for heart regeneration and compact muscle development. We processed their dataset and checked the expression of aEPC markers. As shown in Supplementary Fig. 9c, d, *ptx3a* and *col12a1b* are expressed in the injured 7-day sample but are merely detectable in the uninjured one. *Ptx3a* is primarily expressed in their cluster 2, which was suggested by Sun et al. to give rise to the adjacent clusters, including the *hapln1a*-enriched clusters[13]. This analysis and our results indicate the progenitor property of *ptx3a*⁺ epicardial cells in different heart injury models, and that the pro-regenerative *hapln1a*⁺ epicardial cells are progenies of *ptx3a*⁺ aEPCs. In addition, we previously reported a transcriptomic profile of 31 isolated *tcf21*⁺ cells from uninjured adult hearts[55]. We found that *caveolin-1* (*cav1*), a pan-epicardial marker expressed in all 3 clusters, is required for heart regeneration. In our current dataset, *cav1* expression is broadly observed across clusters, although it is relatively higher in the epithelial and aEPC subpopulations (Supplementary Fig. 4). Similarly, De Bakkers et al. found that deletion of another epicardial gene paired related homeobox 1b (*prrx1b*), blocked heart regeneration with increased fibrosis[14]. In agreement with De Bakkers et al.'s result, we found that *prrx1a* is expressed in both the outermost and inner layers of the epicardium in our dataset, while *prrx1b* expression is much lower and only detected in a few cells (Supplementary Fig. 4). Our re-analyses indicate that *ptx3a*⁺ cells are slightly reduced in the *prrx1b* mutant (Supplementary Fig. 9e–g). Thus, the pro-regenerative *cav1*⁺ or *prrx1*⁺ epicardial cells are broader populations that include the aEPCs and their progenies in the regenerating heart. These analyses define aEPCs as the primary cellular source of essential epicardial cell progenies for heart regeneration in zebrafish.

We have shown that aEPCs express published pro-regenerative factors such as *aldh1a2*, *fn1*, *fstl1*, *tmsb4x*, and *col12a1*[16,34,42,43,56,57] (Supplementary Fig. 4). Expression of these factors is likely reduced upon aEPC ablation, which may contribute to regeneration defects. To further assess how aEPCs support myocardium regeneration, we checked the epicardium-derived mitogenic factor *nrg1*[41]. scRNA-seq data shows that *nrg1* expression is enriched in the mesenchymal epicardial cells (Fig. 3a). We found that aEPC ablation significantly reduced *nrg1* expressing cells in the wound at 7 dpa (41% reduction compared to vehicle treatment and 43% reduction compared to the Mtz-treated NTR-negative group: Fig. 8a, b). In addition, Sun et al. demonstrated that the *hapln1a*⁺ epicardial cells mediate HA deposition for myocardial regeneration[13]. As expected, aEPC ablation significantly reduced the number of *hapln1a*⁺ mesenchymal epicardial cells in the wound by ~42% compared to the control groups at 7 dpa (Fig. 8c, d).

Thus, reduced Nrg1 signals and HA deposition in the aEPC-depleted heart likely contribute to reduced CM proliferation and observed regeneration defects. In summary, our results suggest that aEPCs are the primary cellular source of the essential epicardial cell progenies and paracrine factors required for successful heart regeneration. Nrg1, HA, and *hapln1a*⁺ cells are among the downstream effectors of aEPC activation in supporting heart regeneration.

**Tgfβ signaling regulates aEPC EMT and differentiation**

We next asked how epicardial EMT regulates heart regeneration. The Tgfβ pathway is known as a regulator of EMT and has been reported to play important roles in zebrafish heart regeneration in epicardial cells after heart injury[58]. However, the underlying mechanism is not fully understood. Our scRNA-seq data indicate an injury-induced upregulation of *tgfb1a* in part of the aEPCs, which matches the expression pattern of *snai1a* at 3 dpa (Fig. 9a). HCR staining showed co-expression of *tgfb1a* and *snail1a* in *col12a1b*⁺ aEPCs in the wound at 3 dpa (Fig. 9b). To further test the function of Tgfβ in aEPC EMT, we treated fish with SB431542 (a Tgfβ pathway inhibitor) after amputation injury and assessed heart regeneration at 7 dpa (Fig. 9c)[43]. This treatment led to large blood clots in all hearts (7 of 7 hearts), while DMSO-treated hearts are largely normal with a minor noticeable wound (5 of 6 hearts; Fig. 9d). We observed 55% and 53% reductions of *tcf21*⁺ cells and *ptx3a*⁺ cells, respectively, in the wound after SB431542 treatment. The thickness of the epicardial cell cap that covers the wound is also reduced by 71% on average (Fig. 9e–h). Further HCR staining demonstrated a 68% reduction of *hapln1a*⁺ mesenchymal epicardial cells entering the wound (Fig. 9i, j), suggesting defects of epicardial differentiation and EMT. Thus, Tgfβ inhibition largely mimics the aEPC depletion phenotypes. These results suggest that Tgfβ regulates EMT and mesenchymal cell differentiation of aEPCs, which are essential processes for heart regeneration.

**Comparison with mouse epicardial cells**

To assess the similarities and differences between zebrafish and mouse epicardium, we analyzed a published scRNA-seq dataset of adult mouse epicardial cells (Supplementary Fig. 10). In the injured adult mouse heart, epicardial cells form a multi-cell layer in the wound[59]. Hesse et al. named these cells as epicardial stromal cells (EpiSC) and performed scRNA-seq of FACS-isolated EpiSC 5 days after myocardial infarction (MI)[10]. The dataset comprises 11 clusters that are separated into 3 groups: I, II, and III (Supplementary Fig. 10a). Cells in group I (expressing *Wt1*) are located in the outermost layer of the epicardium. Expression of group III markers are present throughout the activated epicardium but mostly in the inner layers of the epicardium. Group II has both epithelial clusters (expressing *Wt1*) and inner layer clusters and is enriched with ECM-related pathways[10]. We examined the expression of zebrafish cluster markers in the mouse dataset. As shown in Supplementary Fig. 10b–f, the homologs of zebrafish epithelial epicardium markers *Podxl*, *Sema3d*, and *Aldh1a2* are enriched in mouse EpiSC group I. The homologs of zebrafish aEPC markers *Ptx3*, *Col12a1*, *Marcks*, *Lox*, *Hop9Ob1*, *Serpinh1*, and *Tmsb4x* are primarily

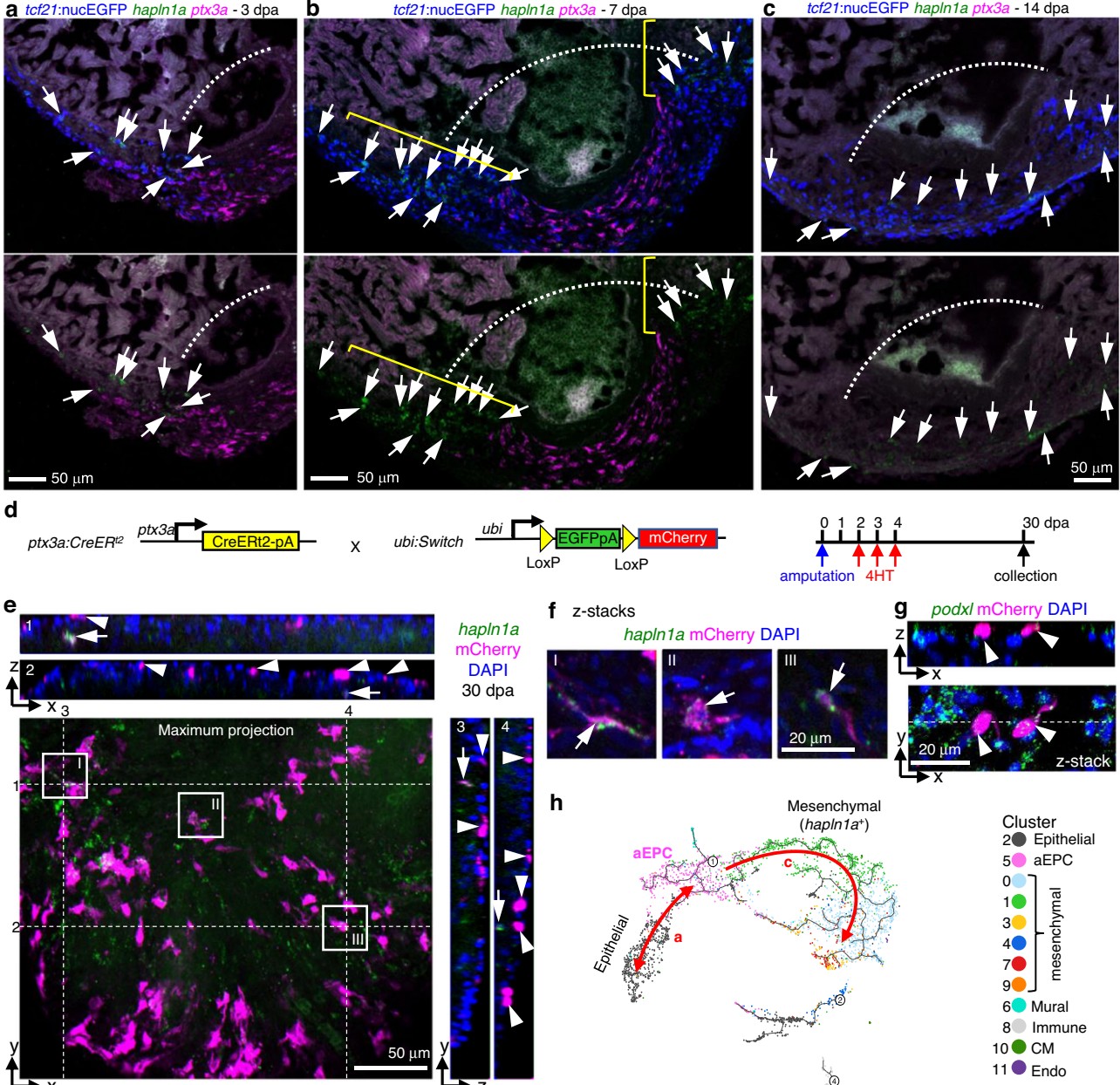

**Fig. 6 | aEPCs give rise to the epithelial and mesenchymal epicardial cells.**
Section images showing HCR staining results of *hapln1a* (green) and *ptx3a* (magenta) at 3 (**a**), 7 (**b**), and 14 dpa (**c**). Epicardial cells are labeled with *tcf21*:nucEGFP (blue). Arrows in **a**–**c** indicate representative *hapln1a*⁺ cells. The brackets in **b** outline the regenerated regions that express *hapln1a*. White dashed lines indicate the injury sites. Scale bars, 50 μm. **d** Schematic of transgenic lines and experimental design to define the fate of aEPCs. 4HT was used at 5 μM. **e** Orthogonal view of a z-stack image showing the ventricular surface layers from hearts carrying the *ubi:Switch;ptx3a:CreER^t2* reporters. A maximum projection image of the x-y plane is shown with *hapln1a* expression (HCR staining) in green and mCherry in magenta. White dashed lines and numbers indicate positions for views of the y-z planes (right) and the x-z planes (top), respectively. Nuclei were stained with DAPI (blue)

and was omitted from the maximum projection image to keep image clarity. Arrows indicate representative *hapln1a*⁺mCherry⁺ cells. Arrowheads indicate mCherry⁺ cells in the epithelial layer. Scale bar, 50 μm. The framed regions are enlarged to show details in **f**. **f** Optical section (z-stack) images of the framed regions in **e**. Arrows indicate *hapln1a*⁺mCherry⁺ cells. Scale bar, 20 μm. **g** Orthogonal view of a z-stack image showing the ventricular surface layers from hearts carrying the *ubi:Switch;ptx3a:CreER^t2* reporters. An optical section image (x-y plane) is shown at the bottom, and the x-z plane of z-stacks in shown on top. mCherry and HCR staining signals of *podxl* are shown in magenta and green, respectively. The white dashed line indicates position for the view of the x-z plane. Arrowheads indicate *podxl*⁺mCherry⁺ epithelial epicardial cells. **h** A UMAP highlights pseudotime trajectories a and c.

expressed in mouse EpiSC group II. The zebrafish mesenchymal or mural epicardium markers *Hapln1*, *Pdgfra*, and *Pdgfrb* are enriched in mouse EpiSC group III. In addition, zebrafish *fn1a*, *psmb1*, and *atp5mc1* are markers for aEPCs, but are also expressed in part of the epithelial epicardium cluster (Fig. 3e and Supplementary Fig. 5). Similarly, the mouse homologs *Fn1*, *Psmb1*, and *Atp5g1* are highly expressed in EpiSC

groups I and II (Supplementary Fig. 10b–f). Thus, mouse EpiSC groups I, II, and III are comparable to zebrafish epithelial, aEPC, and mesenchymal/mural subsets, respectively.

However, unlike in zebrafish, mouse epithelial epicardial cells do not give rise to mesenchymal EpiSCs in the infarct[10]. This observation was also supported by other studies[6,60]. Although group II of the mouse

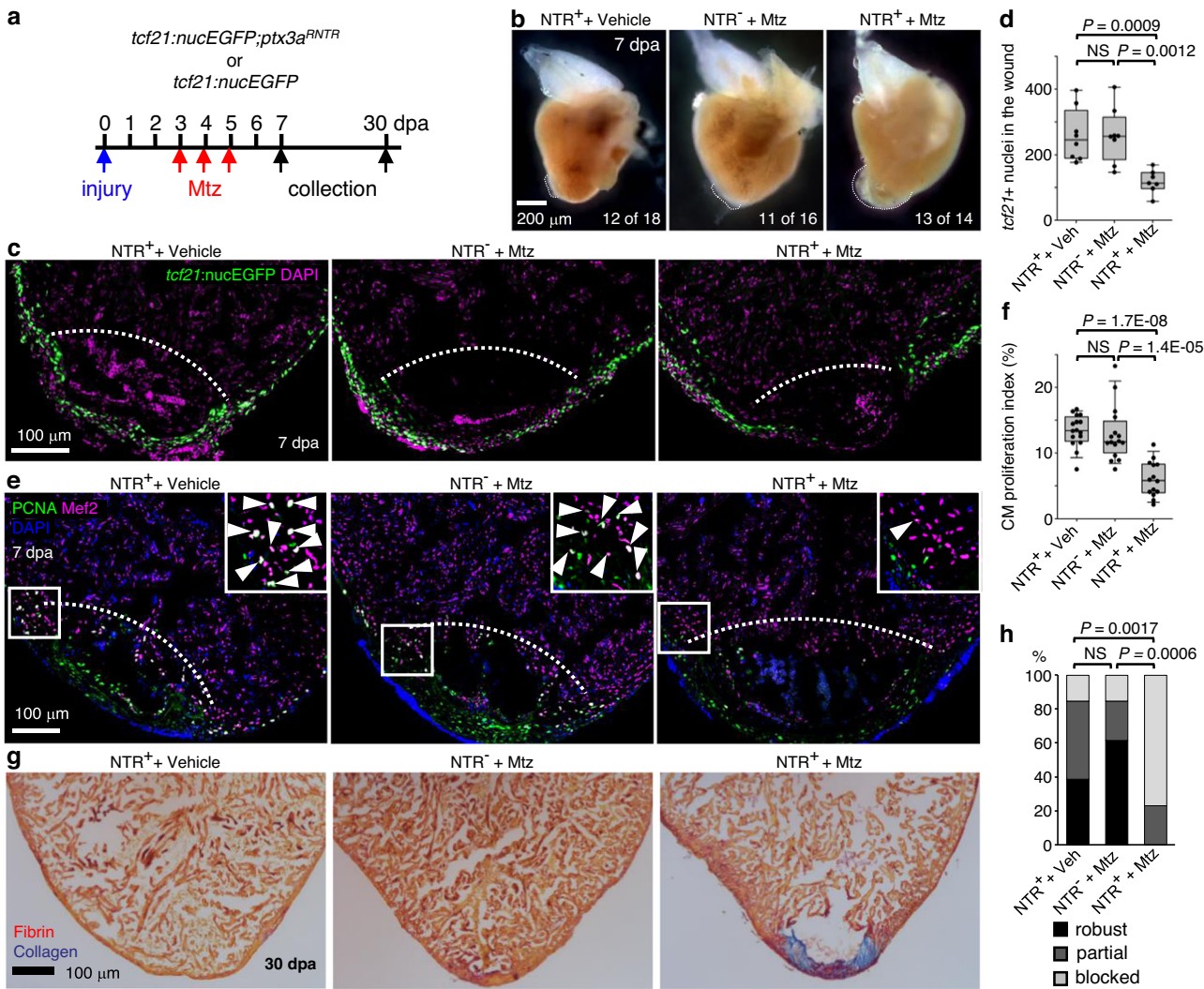

**Fig. 7 | aEPC ablation blocks heart regeneration. a** Experimental design. Siblings carrying the *ptx3a^RNTR* allele and/or the *tcf21:nucEGFP* reporter were treated with 5 mM Mtz or vehicle (Ctrl) from 3 dpa to 5 dpa. **b** Whole mount images of hearts collected at 7 dpa. Dash lines denote the injury sites. Large blood clot and extra tissue were observed in hearts of Mtz treated NTR⁺ animals (13 of 14) but not in those from the vehicle-treated NTR⁺ fish (12 of 18) or Mtz treated NTR⁻ fish (11 of 16). Scale bar, 200 μm. **c** Section images of injured ventricles from 3 treatment groups at 7 dpa. The epicardial cells are labeled with *tcf21:nucEGFP* (green). Nuclei were stained with DAPI (magenta). White dashed lines indicate the injury sites. Scale bar, 100 μm. **d** Quantification of EGFP⁺ cells in the wound region from experiments in **c**. From left to right, *n* = 8, 8, and 7, respectively. NS not significant. Two-tailed Student's *t* test. **e** Section images of injured ventricles from 3 treatment groups at 7 dpa. Ventricular CM proliferation was assessed by anti-PCNA (green) and Mef2

(magenta) staining. Nuclei were stained with DAPI (blue). The framed regions are enlarged to show proliferating CMs (some denoted with arrowheads). **f** Quantified PCNA⁺ CM indices in injury sites in experiments from **e**. From left to right, *n* = 16, 16, and 14, respectively. NS, not significant. Two-tailed Student's *t* test. **g** Section images of ventricles at 30 dpa stained with Acid Fuchsin-Orange G to characterize non-muscle components in the injuries (blue for collagen, red for fibrin). **h** Semiquantitative assessment of cardiac injuries based on muscle and scar morphology (robust, partial, or blocked regeneration). Data were analyzed using Fisher's exact test. *n* = 13 for each treatment group. NS not significant. Box plots show the median (center line), upper and lower quartiles (box limits), minimum and maximum values (whiskers), and individual values (points). Source data are provided as a Source data file.

EpiSC does express markers of zebrafish aEPCs, Hesse et al. study showed no differentiation trajectory from group II to cells in other groups. These differentiation deficiencies may contribute to the limited regenerative capacity of the adult mouse heart, which warrant further genetic studies in mice. In all, this comparison demonstrates both similarities and differences in epicardial populations between zebrafish and mice. It also implies that activating such a progenitor state in mice has the potential to promote cardiac repair.

## Discussion

Here we have defined the epithelial, mesenchymal, and mural subsets of the epicardial lineage in adult zebrafish. We identified the *ptx3a⁺col12a1b⁺* epicardial cells as the adult progenitors - aEPCs. These aEPCs undergo

EMT and orchestrate ECM remodeling and cell differentiation to mediate heart regeneration (Fig. 9k). Thus, augmenting aEPC activation after heart injury is of potential value for enhancing heart regeneration.

Our modRNA-assisted genetic tracing demonstrated an active epicardial EMT process during heart regeneration in adult zebrafish. It was reported that epicardial EMT occurs prior to fate specification in a chick heart development model[61]. Hampering epicardial EMT in mice abolishes epicardial lineages and leads to severe heart development defects[62–64]. In contrast to zebrafish, adult mammalian epicardial cells have no or limited EMT upon heart injuries[6,60]. Our study further highlights that deficiency in epicardial EMT may contribute to the limited regenerative capacity of the adult mammalian heart. In addition, we demonstrate pericardial sac injection of modRNA as an

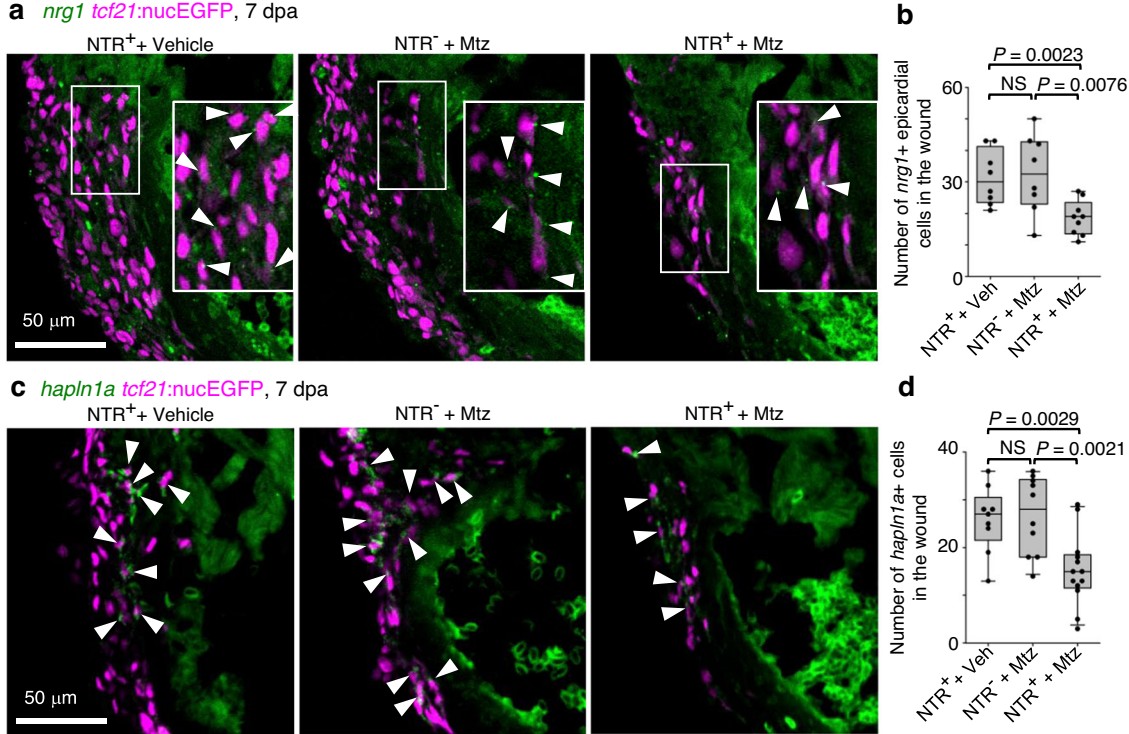

**Fig. 8 | aEPC ablation reduced *nrg1* and *hapln1a* expression in the wound.**
**a** Section images of the injury site showing HCR staining of *nrg1* in green and *tcf21*:nucEGFP in magenta. The framed regions are enlarged to show details with arrowheads denoting representative *nrg1*+EGFP+ cells. Scale bar, 50 μm.
**b** Quantification of *nrg1*+ epicardial cells in the wound regions shown in **a**. From left to right, *n* = 8, 8, and 9, respectively. NS, not significant. Two-tailed Student's *t* test.
**c** Section images of the injury site showing HCR staining of *hapln1a* in green and *tcf21*:nucEGFP in magenta. Arrowheads indicate representative *hapln1a*+EGFP+ cells. Scale bar, 50 μm. **d** Quantification of *hapln1a*+EGFP+ cells in the wound regions shown in **c**. From left to right, *n* = 9, 10, and 13, respectively. NS, not significant. Two-tailed Student's *t* test. Box plots show the median (center line), upper and lower quartiles (box limits), minimum and maximum values (whiskers), and individual values (points). Source data are provided as a Source data file.

approach for mechanistic studies of the epicardium, which can be applied in multiple contexts in both zebrafish and mammals (e.g., gain- and loss-of-function assays for epicardial activation and EMT).

Previous work with the *tcf21:CreER*[t2] transgenic line discovered that epicardial cells contribute to a large amount of perivascular (or mural) cells during zebrafish heart regeneration[18]. However, since the *tcf21* reporters also marks part of the *pdgfrb*+ mural cells during regeneration, it was unclear whether these perivascular contributions originated from the existing *tcf21*:CreER+*pdgfrb*+ mural cells or epicardial progenitors. Our pseudotime analysis uncovers the existence of both trajectories, and we confirm experimentally that at least a subset of the mural cells in the injury site is derived from aEPCs. The *hapln1a*+ mesenchymal subset is divided into multiple clusters, which indicates further heterogeneity. These subsets may have additional functions in supporting regeneration even if their relative proportion remains unchanged. Gene expression profiles suggest that some mesenchymal epicardial cells may have a fibroblast identity, which warrants further investigation. Our GO term analysis suggests a number of unique contributions that each core epicardial cluster makes during regeneration. How these subsets coordinate with each other and with other cardiac cell types to exert efficient regeneration needs further study. A recent study by Sun et al. demonstrated that the *hapln1a*+ epicardial cells mediate HA secretion and myocardial development and regeneration[13]. Thus, both studies suggest that the entire epicardial population actively participated in the regeneration process with diverse cellular and paracrine contributions.

We showed that genetic ablation of *col12a1b*-expressing aEPCs upon heart injury led to the formation of collagen-enriched scar tissues at 30 dpa. This suggests that the initial deposition of the pro-regenerative collagen XII[44,45] may be necessary for regeneration.

Notably, it was reported that transient collagen deposition is required for zebrafish heart regeneration in a cryoinjury model[37]. The pro-regenerative collagen remodeling, in terms of the composition of collagen components and the associated deposition timing warrants further investigation. Although it has been shown that Tgfβ inhibition blocks heart regeneration[58], our results provide further cellular insights that Tgfβ-regulated EMT and differentiation of aEPCs contribute to heart regeneration. Spatiotemporal activity of the Tgfβ signaling may regulate dynamic ECM deposition. Moreover, the function of *ptx3a* implies a major role of aEPCs in mediating inflammation during regeneration[46]. Thus, our discovery suggests aEPCs as a molecular hub connecting ECM remodeling and immune responses during regeneration. Our findings also open new research avenues to precisely manipulate the regeneration program.

In mammals, the adult epicardium shows analogous activation upon heart injury (such as re-activation of embryonic gene expression, proliferation, and secretion), but this activation is limited in terms of mitogen secretion, EMT, and differentiation capacity[6,10,60]. The similarities between zebrafish aEPCs and mouse epicardial-derived cells suggest that awakening the progenitor potential in the adult mammalian epicardium could promote cardiac repair after myocardial infarction. In all, our study has revealed the plasticity of adult epicardial cells and highlighted the aEPCs as a target for enhancing cardiac regeneration.

## Methods

### Animal maintenance and procedures
Animal procedures were approved by the Institutional Animal Care and Use Committee (IACUC) at Weill Cornell Medical College. Adult zebrafish of the Ekkwill and Ekkwill/AB strains were maintained following standard protocols[2,65]. Briefly, water temperature was

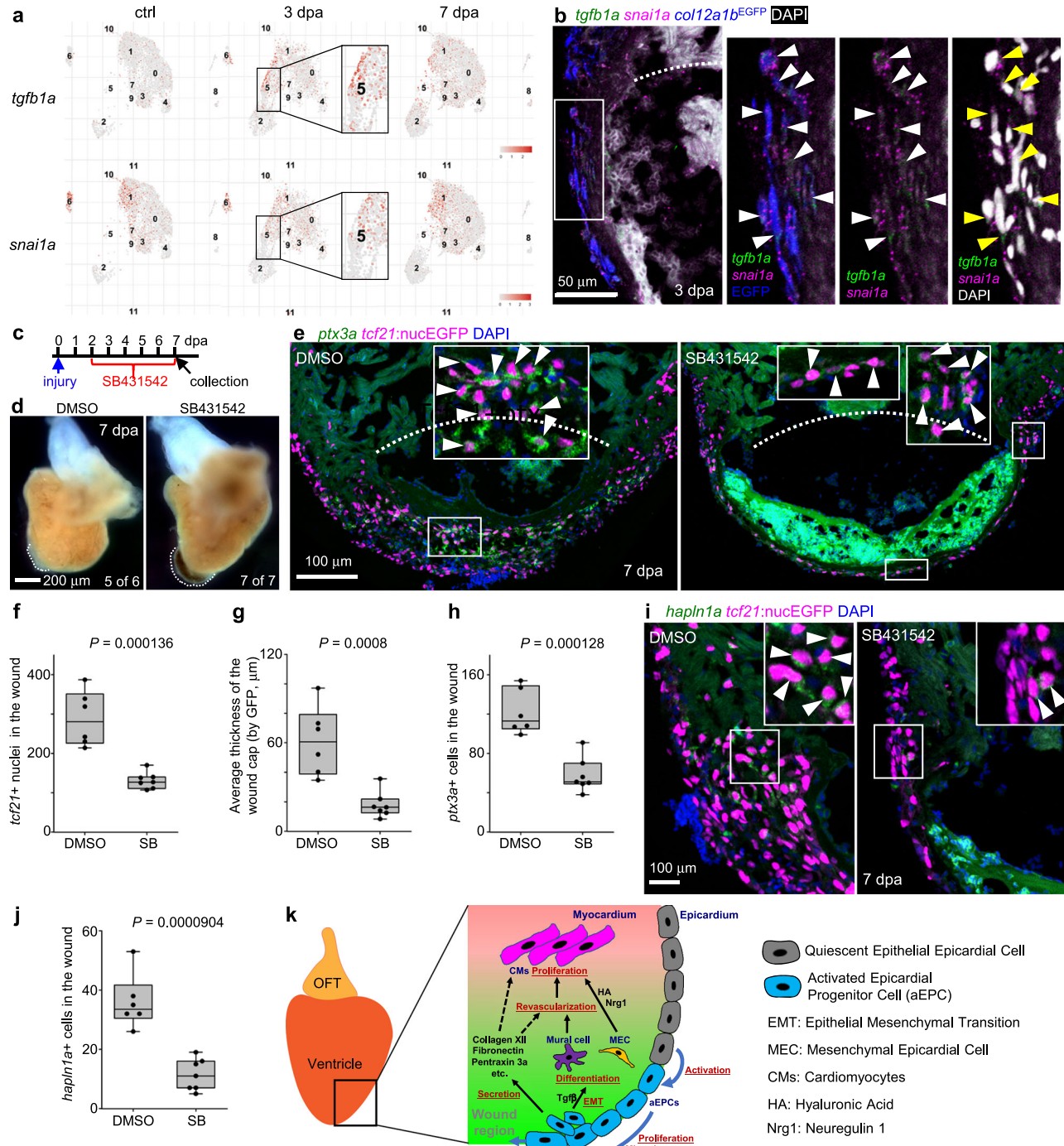

**Fig. 9 | Tgfβ signaling regulates aEPC EMT and differentiation. a** UMAPs showing *tgfb1a* and *snai1a* expression across samples. The aEPC population (cluster 5) is highlighted in frames. **b** Section images of the injury site showing HCR staining signals of *tgfb1a* and *snai1a* at 3 dpa in green and magenta, respectively. *col12a1b*EGFP is shown in blue. The white dash line denotes the injury site. The framed region is enlarged to show details on the right with different channel combinations. DAPI staining is shown in white in the last panel. Arrowheads indicate representative *tgfb1a*+*snai1a*+EGFP+ cells. **c** Experimental design for SB431542 treatment. **d** Whole-mount images of hearts collected at 7 dpa. Dash lines denote the injury sites. Large blood clot and extra tissue were observed in SB431542-treated hearts (7 of 7), but not in those from the DMSO-treated fish (5 of 6). Scale bar, 200 μm. **e** Section images of injured ventricles at 7 dpa. The epicardial cells are labeled with *tcf21*:nucEGFP (magenta) and HCR staining signals of *ptx3a* in green. Nuclei were stained with DAPI (blue). White dashed lines indicate the injury sites. The framed regions are enlarged to show details with arrowheads indicating

representative *ptx3a*+EGFP+ cells. Scale bar, 100 μm. Quantifications of *tcf21*:nucEGFP+ nuclei in the wound region (**f**), average thickness of the epicardial cap covering the wound (**g**), and number of *ptx3a*+ cells (**h**) from experiments in **e**. n = 6 (DMSO) and 7 (SB431542), respectively, for each quantification. Two-tailed Student's *t* test. **i** Section images of the injury site showing HCR staining of *hapln1a* in green and *tcf21*:nucEGFP in magenta. The framed regions are enlarged to show details with arrowheads indicating representative *hapln1a*+EGFP+ cells. Scale bar, 100 μm. **j** Quantification of *hapln1a*+EGFP+ cells in the wound regions shown in **i**. n = 6 (DMSO) and 7 (SB431542), respectively. Two-tailed Student's *t* test. **k** The working model. For simplicity, mesenchymal epicardial cells away from the injury site are omitted. The dashed lines indicate predicted mechanisms. Box plots show the median (center line), upper and lower quartiles (box limits), minimum and maximum values (whiskers), and individual values (points). Source data are provided as a Source data file.

maintained at 28 °C, and fish were kept on a 14/10 light/dark cycle at a density of 5–10 fish per liter. Animals between 3 and 12 months of both sexes were used for adult experiments. Heart resection injury was done following the published protocol[66]. Briefly, zebrafish were anesthetized in 0.02% Tricaine (Pentair, cat# TRS1) and placed ventral side up into a moist, slotted sponge. Iridectomy scissors (World Precision Instruments, cat# 501234 and 14124) are used to make a small incision that penetrates the skin and pericardial sac. The ventricle is exposed by gentle abdominal pressure, and 10–15% of the ventricle at the apex is removed by scissors. After surgery, fish are returned to the fish water and stimulated to breathe by vigorously squirting water over the gills with a pipette. For aEPC ablation, we applied amputation injury and then bathed fish in 5 mM metronidazole (Mtz, Sigma-Aldrich, M1547) for 3 successive days from 3 to 5 dpa with fish water being changed daily[15]. For lineage tracing, embryos and larvae were treated with 10 μM 4-Hydroxytamoxifen (4HT, Sigma-Aldrich, H7904) in fish water for time periods as mentioned in the figures. Adult fish were placed in a mating tank of aquarium water containing 5 or 10 μM 4HT as noted in the figures. Fish were maintained for 16 h, rinsed with fresh aquarium water, and returned to a recirculating aquatic system for 8 h, before repeating this incubation, as noted in the figures. For Tgfβ pathway inhibition, adult fish were incubated with 20 μM SB431542 as noted in the Figures with daily water changes[43]. The following previously published lines were used: *Tg(tcf21:nucEGFP)^pd41* (ref. 18), *Tg(tcf21:H2A-mCherry)^pd252* (ref. 67), *Tg(pdgfrb:EGFP)^ncv22* (ref. 68), and *Tg(ubi:loxP-EGFP-loxP-mCherry)^cz1701* (*ubi:Switch*) (ref. 54). Newly generated lines are described below. All reporters were analyzed as hemizygotes. Transgenic lines generated in this study are readily available on request.

## modRNA synthesis and injection

modRNAs of Cre were synthesized following the published protocol[69]. Briefly, modRNA was transcribed in vitro from a Cre plasmid template using a customized ribonucleotide blend of anti-reverse cap analog CleanCap Reagent AG (3′ OMe) (6 mM, TriLink Biotechnologies, cat# N-7413), guanosine triphosphate (1.5 mM, Life Technology), adenosine triphosphate (7.5 mM, Life Technology), cytidine triphosphate (7.5 mM, Life Technology) and N1- Methylpseudouridine-5′-Triphosphate (7.5 mM, TriLink Biotechnologies, cat# N-1081). mRNA was purified using the Megaclear kit (Life Technology, cat# AM1908) and treated with antarctic phosphatase (New England Biolabs), followed by re-purification using the Megaclear kit. mRNA was precipitated with ethanol and ammonium acetate, and resuspended in 10 mM Tris-HCl, 1 mM EDTA. The Cre ORF sequence is: Atgtccaatttactgaccgtacaccaaaatttgcctgcattaccggtcgatgcaacgagtgatgaggttcgcaagaacctgatggacatgttcagggatcgccaggcgtttctgagcatacctggaaaatgcttctgtccgtttgccggtcgtgggcggcatggtgcaagttgaataaccggaaatggtttcccgcagaacctgaagatgttcgcgattatcttctatatcttcaggcgcgcggtctggcagtaaaaaactatccagcaacatttggccagctaaacatgcttcatcgtcggtccgggctgccacgaccaagtgacagcaatgctgtttcactggttatgcggcggatccgaaaagaaaacgttgatgccggtgaacgtgcaaaacaggctctagcgttcgaacgcactgatttcgaccaggttcgttcactcatggaaaatagcgatcgctgccaggatatacgtaatctggcatttctggggattgcttataacaccctgttacgtatagccgaaattgccaggatcagggttaaagatatctcacgtactgacggtgggagaatgttaatccatattggcagaacgaaaacgctggttagcaccgcaggtgtagagaaggcacttagcctggggggtaactaaactggtcgagcgatgggatttccgtctctggtgtagctgatgatccgaataactacctgtttttgccgggtcagaaaaaatggtgttgccgcgccatctgccaccagccagctatcaactcgcgccctggaaggggattttgaagcaactcatcgattgatttacggcgctaaggatgactctggtcagagatacctggcctggtctggacacagtgcccgtgtcggagccgcgcgagatatggcccgcgctggagtttcaataccggagatcatgcaagctggtggctggaccaatgtaaatattgtcatgaactatatccgtaacctggatagtgaaacaggggcaatggtgcgcctgctagaagatggcgattag. Up to 2 μl of 10 mg/ml purified Cre modRNA was injected into the pericardial sac of each fish following the published protocol[22]. Briefly, zebrafish were anesthetized in 0.02% Tricaine (Pentair, cat# TRS1) and placed ventral side up into a moist, slotted sponge. Under the stereomicroscope, insert the needle tip of a 10 μl Hamilton syringe (vwr, cat# 80008) at a 40-degree angle relative to the body axis without poking the heart. Once the needle tip is inside the pericardial sac, gently inject 2 μl of the modRNA solution.

## Generation of knock-in lines

We applied a Crispr/Cas9 knock-in strategy to insert EGFP-poly A, or mScarlet-P2A-NTR-poly A cassette right after the start codon following a published protocol[70]. Briefly, 2-3 sgRNAs close to the ATG site were selected by using a design tool from Integrated DNA Technologies, Inc. HDR (Homology-directed repair) templates were designed to include a mutated sgRNA target site, homology arms (HAs) flanking the ATG start codon, and a cassette encoding EGFP, or mScarlet-P2A-NTR with polyA. Gene-specific Alt-R crRNAs were synthesized by IDT. Bipartite synthetic sgRNAs were heteroduplexed by using crRNAs and a tracrRNA according to manufacturer recommendations. HDR sequences were synthesized by GENEWIZ and ligated to a pUC57 vector. HDR templates were digested from the pUC57 vector with flanked blunt-ends restriction enzymes and were column-purified. 250 ng/μl sgRNA, 250 ng/μl rCas9 (PNA Bio), and 50 ng/μl HDR templates were injected into one-cell stage embryos. Stable transgenic lines with seamless insertion alleles were identified by genotyping and sequencing of F1s. The following knock-in alleles were generated for this study:

***Tg(ptx3a:mScarlet-P2A-NTR).*** A mScarlet-P2A-NTR-poly A cassette was inserted right after the start codon using the following gRNAs and HAs. sgRNA1: AACAAGGAGGACCATCCAAG; sgRNA2: GTCTTGCTCATATTAACACA. 5′HA: 5′-agtACTTGCATTTAATACAGATATTCCCAAAATGTTTCCAGAGCACTATTTCAAAATTCATTGCTATTTCCAGGTTTCCTTACATCTTTCCCAACATCCTCCCCCAAAATCTCTCTCTTTCTCTCTATGCCTGTGCTGCAGGAGCTGTACAGCAATATGACTACCCAAACTACCTCCAACGAAGAATTTCTCACTGGAGAAACTCACTCTCTCCCCCACCACTCTCTCTCACTCTCTCTCTCTCTCTCTCTCACTCTCTCTCTCTCTCTCTCTCTCTCTCTCTCTCTCTCAGCGCAATGTTTAAATAGACCACAGAGCTGCTCTATCAGGCTCAGTTGCCTTTCAGCTGCAGTTCACCAAACAAGCAGGAGTCCAGCCAACAAATTGCTCAGCATCTAtCAtGGtGGtCCAaCCAtGAcGACAAAGACAATTATACACAAAAGCTGAGTgTTcCaCAaATatACtC-3′; 3′HA: 5′-agtACTTATTGCATAC TCAAAACACAATAGTCAAATACAGTAATCAATCAGTTATTCTAATTTACATCATATCATGCATCCTGTTAAACTACAAGAAATATTCCAGCATTTAACTCTTAACTTTTAACTTTTAACATAACTCCTTTCTGAAAGGTAGTGGAATTATACATCTCTTTATCTCACCTCTTACTCTGATCACAGTTGAGTATTTCTTTCTGGTTACTCACCTTCCAACTGGTCCTCATCAGGTATTTCATTAAAATAATTGTGTCCAAAGCCCACTTCGATATCATCGTCATAATTGTCAGATACCGCCATGGACAATGAAGCCACCAGACACATGGTCTGGAGGAATGTCCGCATGGACTTGCTGGC-3′. The underlined sequences are the mutated gRNA binding sites with the mutations shown in lower case letters. The full name of this line is *Tg(ptx3a:mScarlet-P2A-NTR)^wcm107* (*ptx3a^RNTR* for short).

***Tg(col12a1b:EGFP).*** An EGFP-poly A cassette were inserted right after the start codon using the following gRNAs and HAs. sgRNA1: CCTGACCGACATCTTCACCC; sgRNA2: CTGACATCAGTGAGAGCCCA, sgRNA3: AATTATTAGAGTCTTAACCA; 5′HA: 5′-cccGGGTTTTACAGTGCATGTAAAGTCTGAATTTGTCCAGGCACAACAGGGGGCGCAGAAGCCCTGCAGATGTTTTTCCATCCCTACCATGCAGGATCCTAAACCAGGCAGCTTTCCAGACAATCCCATATAGTCTATGAACTCACAGTTTCCAATGTTAAACTGATGAATCAGCAGGGTTTGCGATGACATAAAGCATATGCAGCCTATAATTGTTTTTACAGTTGAACTAAAATGTTCTTTCTAATAACCGCACAGTGAATTATTTTAATTAATCCCCCTTTTATTAATATTTTCCAAAAaTAcTtGtGaCTTtACgAAcGAAAAAAAAAAACTTTTCTTCTCTGACCAAAGAAGCAGCTGGGATTTTGTATGAAAATATAATTAAAATCAGATTTAATTGAGATCTCTCAATGCTGGAGTGGAGCATGTATTTTCCCCTTGATTTGTGAGCTGTGTGTGTGTGTGTGTGTGTGTGTGTGTGTGTTTGTGTGTTTGTGTGTCTGTGTGTAATCTGCTTGAATTTTCAGATTTGATTGATGTTCTGCTGATACCGGGGTGAAG−3′; 3′HA: 5′-gatATCAATGCACATGAGGAATT

ATTTGGCTCCAGTCCTTTTGTTCATTCCTTGAACAAAGGCAATGAAAA
ATAATGTAAGCACTTAATATAATGGGCATCTATTACAGTAAAGTCGA
TATTTATAGAGTTTAAAACAGAAATCAGACTATGATCTAAACTGTTGG
AAGAAAGTGGTTTAATTTACTGTAATTGGATGAATGGAAAGGCATTTG
TTGGCTTTTTTTTTTTTTTTTTACgTTGtGCcCgaACgGAaGTCAGTGCTG
CGAGTGTAAAGAGAGCCACCGCTGCCAAATGCCgcACtGA-3. The underlined sequences are the mutated gRNA binding sites with the mutations shown in lower case letters. The line in use was generated by using sgRNA1. The full name of this line is *Tg(col12a1b:EGFP)*[wcm108] (*col12a1b*[EGFP] for short).

## Generation of *Tg(ptx3a:CreER*[t2]*)* zebrafish

The translational start codon of *ptx3a* in the BAC clone CH211-272F3 was replaced with a *CreER*[t2]*-polyA* cassette in SW105 cells using recombineering[71,72]. The 5′ and 3′ HAs for recombination are 300–500-base pair (bp) fragments upstream and downstream of the start codon and were PCR amplified with restriction enzyme site and inserted into a vector to flank the *CreER*[t2]*-polyA* cassette. The primer sequences were: ptx3a-5HA-forward: gcggccgcAGTACTTGCATTTAATACAGAT; ptx3a-5HA-reverse: gaattcGTGTTAATATGAGCAAGACTCA; ptx3a-3HA-forward: gggcccAGtACTTATTGCATACTCAAAAC; and ptx3a-3HA-reverse: GCCAGCAAGTCCATGCG. The same technology was used to replace the loxP site in the BAC vector with a cassette containing the Tol2 fragments, as well as a lens-specific crystallin promoter upstream of mCherry (iTol2Amp-γ-crystallin:RFP, a gift from Nadia Mercader Huber; Addgene plasmid # 108455)[73]. The final BAC was purified with Nucleobond BAC 100 kit (Clontech) and co-injected with 50 ng/µl Tol2 RNA into one-cell-stage zebrafish embryos. Stable transgenic lines were selected. The full name of this line is *Tg(ptx3a:CreER*[t2]*)*[wcm109].

## Generation of *Tg(tcf21:loxP-BFP-pA-loxP-mCherry-NTR)* zebrafish

To make a BAC construct, the translational start codon of *tcf21* in the BAC clone DKEYP-79F12 was replaced with the *loxP-BFP-polyA-loxP-mCherry-NTR-polyA* cassette by Red/ET recombineering technology (Gene Bridges)[74]. The 5′ and 3′ homologous arms for recombination were a 50-base pair (bp) fragment upstream and downstream of the start codon and were included in PCR primers to flank the *loxP-BFP-polyA-loxP-mCherry-NTR-polyA* cassette. To avoid aberrant recombination between the insertion cassette and the endogenous *loxP* site in the BAC vector, we replaced the vector-derived *loxP* site with an I-Sce I site using the same technology. The final BAC was purified with Nucleobond BAC 100 kit (Clontech) and co-injected with I-Sce I into one-cell-stage zebrafish embryos. Stable transgenic lines with bright fluorescence were selected. The full name of this line is *Tg(tcf21:loxP-BFP-pA-loxP-mCherry-NTR)*[wcm110] (*tcf21:BsRNTR* for short).

## Single cell-RNA sequencing

Ventricles were collected from adult *tcf21:nucEGFP* fish at 6 months of age. Ventricular nucEGFP+ epicardial cells were isolated following the published protocol[55]. Briefly, ventricles were collected on ice and washed several times to remove blood cells. Ventricles were digested in an Eppendorf tube with 0.5 ml HBSS plus 0.13 U/ml Liberase DH (Roche) at 37 °C, while stirring gently with a Spinbar® magnetic stirring bar (Bel-Art Products). Supernatants were collected every 5 min and neutralized with sheep serum. Dissociated cells were spun down and re-suspended in DMEM plus 10% fetal bovine serum (FBS) medium with 1.5 µg/ml propidium iodide (PI) and sorted using a Becton-Dickinson Aria II sorter for EGFP-positive and PI-negative cells. The isolated cells were sent to the Epigenomics Core Facility of Weill Cornell Medicine for single cell RNA-seq library preparation using the 10x Genomics Chromium Single Cell 3′ GEM, Library & Gel Bead Kit v3, and Chromium Single Cell B Chip Kit. The libraries were sequenced on a pair-end flow cell with a 2 ×50 cycles kit on Illumina HiSeq4000.

## scRNA-seq analysis

The raw reads were aligned and processed with the CellRanger pipeline (v. 3.0.2) using the zebrafish genome version GRCz10. Subsequent analyses were performed in R (v. 3.5.1) following the recommendations of Amezquita et al. (https://osca.bioconductor.org/)[75] using numerous functions provided in the R packages scater (v. 1.10.1) and scran (v. 1.16.0)[76,77] as well as Seurat (v. 3.1.0) following the tutorials of the Satija Lab (https://satijalab.org/seurat/)[78]. We first removed low-quality droplets and rarely covered genes from all samples: cells were required to have a minimum of $10^{2.5}$ genes and maximum of 5% mitochondrial reads; and genes were removed if they were detected in either fewer than 0.2% of the cells or in fewer than 5 cells per sample. Read counts of 3 samples were normalized using *SCTransform* as implemented in Seurat v3.1 correcting for batch effects between the samples[79]. For visualizations and additional downstream analyses, the SCTransform-normalized (log-transformed) expression values were used unless noted otherwise.

For identifying clusters of cells with similar global transcriptomes, a shared nearest neighbor graph was constructed using Seurat's *FindNeighbors* function with default settings (e.g. $k = 20$) and using the first 20 principal components following PCA. Clusters were identified with Seurat's *FindClusters* function with the resolution parameter set to 0.3 (ref. [80]). In addition, UMAP coordinates were calculated[81]. We assessed the identity of the cells within the resulting clusters using marker genes detected by Seurat's *FindAllMarkers* function with default settings as well as marker genes identified by SC3 (v. 1.10.1)[82]. Gene Ontology (GO) analysis was performed by using *clusterProfiler* (v. 3.10.1)[83]. By focusing on the core clusters, we re-did all processing steps including removal of genes that were expressed in fewer than 5 cells, calculation of normalized expression values correcting for the batch effect of the different conditions, PCA, clustering, and UMAP calculation.

To infer the developmental order of certain subpopulations within the regenerating samples, we applied the trajectory reconstruction algorithm Monocle 3 (v. 0.1.3)[53]. We converted the read counts into a monocle object and re-processed the data using *preprocess_cds* with the number of dimensions set to 100. Cells were clustered with *cluster_cells* using the UMAP dimensions. To identify the trajectories of individual cells through the UMAP space, *learn_graph* was used. To determine pseudotime values, root nodes were identified for each partition (as determined in the previous step) and pseudotime values were calculated based on each cell's projection on the principal graph.

## Histology and microscopy

Freshly collected hearts were fixed with 4% paraformaldehyde (PFA) for 2 h at room temperature or overnight at 4 °C. Fixed hearts were mounted with Fluoromount G (Southern Biotechnology, cat#0100-01) between two coverslips for imaging of both ventricular surfaces, embedded in low-melting point agarose for whole-mount imaging, or applied to cryosection at a 10 µm thickness. Hybridization Chain Reaction (HCR 3.0) staining of whole-mounted hearts or cryosections was done following the published protocols[84]. HCR probes for *ptx3a*, *col12a1b*, *pdgfrb*, *hapln1a*, *podxl*, *atp5mc1*, *hmgb2b*, *hsp90b1*, *loxa*, *psmb1*, *serpinh1a*, *nrg1*, *snai1a*, and *tgfb1a* were synthesized by Molecular Instruments Inc. For immunostaining of whole-mounted hearts or heart sections[15,67], samples were blocked with 2% bovine serum albumin (BSA, VWR, cat#97061), 1% DMSO, 0.5% goat serum (Thermo Fisher, cat#16210) and 0.5% Triton X-100 in PBS for 1 h at room temperature (RT). Primary antibodies were diluted in the blocking buffer and incubated with hearts overnight at 4 °C. Hearts were then washed with PBS plus 0.1% Tween 20 and incubated with the secondary antibody diluted in the blocking buffer for 1.5 h at room temperature. Hearts were stained with DAPI (Thermo Fisher, D3571) to visualize nuclei. Primary antibodies used in this study include rabbit anti-Mef2

(this study, 1:200), mouse anti-PCNA (Sigma, P8825; 1:350), rabbit anti-Aldh1a2 (GeneTex, GTX124302; 1:350), rabbit anti-DsRed (Takara, 632496; 1:200), and mouse anti-Tnnt (ThermoFisher, MS-295-PABX; 1:100). The Mef2 antibody was generated using a Mef2aa peptide (amino acid 314-512 of XP_021323249.1) as an antigen and is now commercially available at Boster Bio (DZ01398-1). Secondary antibodies (ThermoFisher, 1:200) used in this study were Alexa Fluor 488 goat anti-rabbit and goat anti-mouse, Alexa Fluor 546 goat anti-rabbit and goat anti-mouse, and Alexa Fluor 633 goat anti-mouse. Acid Fuchsin-Orange G staining was performed as described[2]. All antibodies are commercially available and are validated by suppliers.

Bright-field images of whole-mounted hearts were captured using a Zeiss Axiozoom V16 microscope (Zen 2.5 blue edition software). Fluorescent images of whole-mounted and sectioned heart tissues were imaged used a Zeiss 800 confocal microscope (Zen 2.6 blue edition software). AFOG staining images were captured on a Leica Dmi8 compound microscope (Leica Application Suite X software). Analyses of CM proliferation were performed as previously described by counting Mef2 and PCNA nuclei in wound sites[56]. Fluorescence images were exported from Zeiss Zen Blue software and processed in Adobe Photoshop 2022 and ImageJ (v2.0.0-rc-69/1.52p).

## Statistics and reproducibility
Clutchmates, or hearts collected from clutchmates, were randomized into different groups for each treatment. No animal or sample was excluded from the analysis unless the animal died during the procedure. All experiments were performed successfully with similar results from at least 2 biological replicates. For all fluorescent images showing representative results, at least 6 zebrafish were analyzed with consistent results. Sample sizes were chosen based on previous publications[13,15,34] and experiment types and are indicated in each figure legend. All measurements were taken from distinct samples. All statistical values are displayed as Mean +/−Standard Deviation (s.d.). Sample sizes, statistical tests, and $P$ values are indicated in the figures or the legends. All box plots show the median (center line), upper and lower quartiles (box limits), minimum and maximum values (whiskers), and individual values (points). Student's $t$ tests (two-tailed) were applied when normality and equal variance tests were passed. The Mann–Whitney Rank Sum test was used when these failed. Fisher's exact test was used where appropriate. Source data are provided with this paper.

## Reporting summary
Further information on research design is available in the Nature Portfolio Reporting Summary linked to this article.

## Data availability
The scRNA-seq datasets generated in this study have been deposited at NCBI's Gene Expression Omnibus under accession numbers GSE202836. Source data are provided with this paper.

## Code availability
All scripts as well as the code and cell labels used for generating the scRNA-seq based figures can be found at https://github.com/abcwcm/Cao_Epicardium and zenodo[85].

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

## Acknowledgements

We thank Adedeji A. Afolalu, Chaim Shapiro, Soji Hosten, and Chelsea Quaies for fish care, Naoki Mochizuki for the *Tg(pdgfrb:EGFP)^{ncv22}* line, and Geoffrey Pitt, Junsu Kang, and Todd Evans for comments on the manuscript. This work was supported by Rudin Foundation fellowships to Y.X. and J.Y., a predoctoral training grant position (T32-HD060600) and a predoctoral fellowship (F31-HL158168) from National Institutes of Health (NIH) to S.D., a predoctoral training grant position in Stem Cell Biology and Regenerative Medicine from New York State Stem Cell Science program (NYSTEM) to B.P., American Heart Association (AHA) Career Development Award (AHA941434) to M.R.M.H., NIH grants (R01HL142768 and R01HL149137) to L.Z., AHA Career Development Award (18CDA34110108), Weill Cornell Start-up fund, and NIH grant (R01HL155607) to J.C.

## Author contributions

Conceptualization, J.C.; Methodology, F.D., P.Z., M.R.M.H., L.Z., D.B. and J.C.; Investigation, Y.X., S.D., B.P., M.Q, J.Y., Y.C., F.D. and P.Z.; Resources, D.B., L.Z. and J.C.; Writing and editing, Y.X., S.D. and J.C.; Funding Acquisition, S.D., M.R.M.H., L.Z. and J.C.

## Competing interests

The authors declare no competing interests.
