## [Peer Review File · Nature Communications]

Activation of a transient progenitor state in the epicardium is required for zebrafish heart regenerationEditorial Note: Parts of this Peer Review File have been redacted as indicated to remove third-party material where no permission to publish could be obtained.

REVIEWER COMMENTS

Reviewer #1 (Remarks to the Author):

The single cell RNAseq approach was previously used to determine the gene expression profile in epicardium of zebrafish embryos (Weinberger et al), uninjured adult heart (Cao et al), and cryoinjured adult heart (de Bakker et al. 2021). The manuscript added additional information on the roles of epicardium in zebrafish heart regeneration after amputation. The authors identified an epicardial progenitor cell population and demonstrated that these epicardial cells are essential for zebrafish heart regeneration. With cellular labeling, lineage tracing, and cell ablation experiments the authors established that the activated epicardial progenitor cells (aEPCs) are transient and appear only in the injured hearts and are mostly localized in the injury area. aEPCs are multipotent and can undergo epithelial to mesenchymal transition and differentiate into mural cells, epicardial epithelial cells, and epicardial-derived mesenchymal cells. Besides this, they described markers for epicardial, epithelial, and mesenchymal layers. The manuscript is nicely written, easy to follow and well organized. Innovative technologies including modified RNA injection and CRISPR to generate new knock-in transgenic lines were used and their data are of high quality. However, the following should be addressed to clarify the conclusion of this manuscript.

1. The authors use Tcf21 as a pan-epicardium marker. However, their images show that Tcf21 is only expressed in a subset of epicardium. It is also demonstrated by Weinberger et al that Tcf21 shows heterogeneous expression and is distinct from Tbx18 and Wt1b in epicardium. The authors can just use Tcf21 as an epicardial marker.

2. The authors used Pdgfrb as a well-established mural cell marker. They further described mural cells as Tcf21+Pdgfrb+ cells. This is quite confusing. Does this mean that TCF21 is a mural cell marker? In fact, Tcf21 is also a well known fibroblast marker (Acharya et al. 2012, Kanisicak 2016). Perdurance of nuclear fluorescent protein (tcf21:H2R) could be a reason why most mural cells around the vessels show a lower level of Tcf21 reporter signal where most likely Tcf21 is not expressed.

Although scRNAseq data from sorted Tcf21:nucEGFP cells, (Fig. 2H) mural cell cluster shows moderate Tcf21 expression along with different mural cell and smooth muscle marker expressions (Fig. S2C). One noticeable information here is that Tcf21 expresses almost in all cell types. The Tcf21 UMI/mRNA level in the mural cell cluster (cluster 6) and in the cardiomyocyte cluster (cluster 10) seems comparable (Fig. S2A, S2E). Also, in differentially expressed gene list in cluster 6 (mural cells), Tcf21 has negative logFold change.

The author should put less emphasis on Tcf21 expression in mural cells. The Tcf21 expression in the mural cells could be a spillage from the epicardial cells (the dominant FACS sorted population) as it happened to all other cell types (e.g. cardiomyocytes) sorted with it.

3. Reduced Tcf21:nucEGFP signal in Tcf21+Ptx3a+ cells (Fig.4C) and in scRNAseq data in (Fig. 2H, 3B) may not be sufficient to establish "dedifferentiated" state. It can also mean these cells are becoming some other cell type (e.g., fibroblasts, according to recent Col12a1a publication, Bo Hu et al., Nature Genetics, 2022) or just the new progeny cells from epicardial cell proliferation/expansion in the injured area. The finding by Bo et al. took away some novelty from this manuscript although the authors claim that Ptx3a+Col12a1b+ cellular population as multipotent progenitor cells. However, it may be essential to establish aEPC is a progenitor population by other markers.

4. To validate the pseudotime analysis and demonstrate aEPC differentiation into epithelial epicardial cells and mesenchymal (inner) epicardial cells, they showed Ptx3a lineage traced cellular position (epithelial vs inner) and Hapln1a expression. While cellular position and Hapln1a expression clearly demonstrate Ptx3a+ cells differentiation into epicardial mesenchymal cells they also should do a Podxl HCR staining to show differentiation into epithelial epicardial cells.

5. In Fig. 4B, the authors showed at 1dpa (and also 2dpa, Fig S6B), the Tcf21+ epithelial epicardial layer of the entire ventricle express Ptx3a. For cell ablation they started respective treatment from 2 dpa. This makes the interpretation a bit confusing.

The authors characterized molecular signature of aEPC population from the cluster 5 cells which is specific for 3dpa sample. At 3dpa Ptx3a+ cells are confined surrounding the injured area (Fig. 4C). 3 dpa Ptx3a+ population could be more specific than pan-epicardial injury responsive Ptx3a+ population at day 1 and 2 which are from epithelial epicardial cells.

Thus broader epithelial epicardial population (beyond aEPC = cluster 5 cells) could be ablated by 2-4 dpa Mtz treatment (and would show stronger effect by more cell ablation). To follow scRNAseq data, ideally it should be 3-5 dpa MtZ treatment.

Minor correction:

1. In Fig. 7 legend the model should be I (mentioned as H).
2. In the Fig. 2, 4 ,7 the panel orders are a bit random and thus difficult to follow.

Reviewer #2 (Remarks to the Author):

This well conducted and well written study identified a transient progenitor cell population (pdgfa+hapln1a+ mesenchymal epicardial cells) which drives zebrafish heart regeneration. Unfortunately, very recently a paper was published by the Poss group demonstrating similar results: hapln1 defines an epicardial cell subpopulation required for cardiomyocyte expansion during heart morphogenesis and regeneration. This paper is briefly mentioned at the end of the discussion, however, it considerably limits the news value of the present manuscript. The summary highlights aEPCs as a potential target for enhancing cardiac repair. This issue, however, is not further elaborated on in the discussion section. It also remains unclear whether the findings reported in zebra fish can be translated to mice and possibly to humans. scRNAseq on mice epicardial cells have been reported last year (Hesse et al 2021) and it would have been interesting compare the zebrafish data with the available mouse data set.

Minor:

p.4, l.61 : The argument that WT1 is also expressed in other non-epicardial cells also holds for tcf21, certainly in mice. Fig 1 shows the tcf21 distribution in the epicardium; does it also label cardiac fibroblasts and/or endothelial cell in the deeper layers of the myocardium ?

Fig. 2 B-E : cluster numbers are difficult to read when background color is dark; perhaps change to white.

Fig. 2 H : tcf21 also labels cluster 10 which are most likely cardiomyocytes. Could this be due to cell doublets which have not been excluded for the final analysis.

Fig. 7: 5 mM metronidazole is a pretty high concentration and one wonders what the appropriate controls are, a structural derivative of Mtz without biological activity ? In clinical medicine the use of metronidazole has many side effects! What was the effect of the "control" (Ctrl) on normal heart regeneration?

Reviewer #3 (Remarks to the Author):

In this manuscript the authors investigated the different cell types derived from the epicardium and

how these behave and contribute to the regenerative response during zebrafish heart regeneration. To characterize the heterogeneous population of epicardial-derived cells (EPDCs), they performed scRNAseq on uninjured and injured hearts in which EPDCs are GFP labeled by *tcf21* expression and clustered the cells based on their transcriptomes. 8 Main clusters with EPDCs could be identified from the data. The authors focused on cluster 5 based on the expression of pro-regenerative genes (*ptx3*, *col12a1b*). Pseudotime analysis indicates that the cluster5 cells are derived from epithelial epicardial cell state and gives rise to a mesenchymal cell state, suggesting that these cells undergo EMT. Next they generated a transgenic *ptx3:Cre* line and concluded from this that *ptx3* expression precedes the expression of *hapln1a*, confirming the pseudotime results. Finally the authors ablate the *ptx3* expressing cells from the injured heart and observed a decrease in regenerative capacity indicating that *ptx3* expressing cells are important during zebrafish heart regeneration.

My main concern is that this manuscript lacks novelty. The here described heterogeneity of the epicardium and epicardial-derived cells during heart regeneration has been studied extensively and has been well established. In addition, several studies (including studies from this group) have been published already that describe scRNAseq data on these epicardial and epicardial-derived cells and characterize epicardial derived sub-populations (PMID: 26657776, pmid: 34486669, PMID: 35652354, pmid: 35088848). Furthermore, from these and other studies it has become very clear that one or several pro-regenerative subpopulation(s) are required for the regenerative response (PMID: 29610343). What is needed now and is missing from this study is a better understanding of the mechanisms by which these pro-regenerative cells stimulate cardiac regeneration.

Specific points to the authors:

1. The authors injected ModRNA encoding Cre into pericardiac cavity. Authors claim that this will only target the epicardial cells on the heart surface. However, the original protocol for pericardiac injection by Bise and Jazwinska shows that chemicals injected in this way can also enter the underlying myocardium. Authors should substantiate their claims that they target only the surface epicardium and not any cells deeper into the tissue with convincing evidence or adjust their claims.
2. The scRNAseq data. The authors performed scRNAseq on a large number of cells at different conditions and identified different cell clusters. For a better comparison and understanding it would be important to compare these data with previously published scRNA seq data (PMID: 26657776, pmid: 34486669, PMID: 35652354, pmid: 35088848). The authors would need to reanalyse some of the published scRNAseq data and show for example how *ptx3* expression is distributed in these published data and discuss how their *ptx3* pro-regenerative subpopulation relates to the previously identified pro-regenerative subpopulations.
3. The trajectory reconstruction that is applied to the scRNAseq data is very difficult to follow. It is not clear why the authors did not use RNA velocity which is more commonly used to analyse directional changes in cell states.
4. The mechanism by which cluster5 EPDCs drive cardiac regenerations remains unaddressed. The authors suggest that these cells undergo EMT, but they have not analysed this further. The observation that *snail1* is expressed is suggestive but *snail1* expression is also expressed cluster 1 cells. Furthermore, what are the factors that induce the EMT and how does the EMT relate to heart regeneration. Is there less CM proliferation in the *ptx3* ablation experiment and if so why?.

Here, we list each suggestion of the reviewers and describe how we have addressed the suggestions in our revision.

Reviewer #1 (Remarks to the Author):

The single cell RNAseq approach was previously used to determine the gene expression profile in epicardium of zebrafish embryos (Weinberger et al), uninjured adult heart (Cao et al), and cryoinjured adult heart (de Bakker et al. 2021). The manuscript added additional information on the roles of epicardium in zebrafish heart regeneration after amputation. The authors identified an epicardial progenitor cell population and demonstrated that these epicardial cells are essential for zebrafish heart regeneration. With cellular labeling, lineage tracing, and cell ablation experiments the authors established that the activated epicardial progenitor cells (aEPCs) are transient and appear only in the injured hearts and are mostly localized in the injury area. aEPCs are multipotent and can undergo epithelial to mesenchymal transition and differentiate into mural cells, epicardial epithelial cells, and epicardial-derived mesenchymal cells. Besides this, they described markers for epicardial, epithelial, and mesenchymal layers. The manuscript is nicely written, easy to follow and well organized. Innovative technologies including modified RNA injection and CRISPR to generate new knock-in transgenic lines were used and their data are of high quality. However, the following should be addressed to clarify the conclusion of this manuscript.

We thank the reviewer for the effort to review our manuscript and the supportive comments.

1. The authors use Tcf21 as a pan-epicardium marker. However, their images show that Tcf21 is only expressed in a subset of epicardium. It is also demonstrated by Weinberger et al that Tcf21 shows heterogeneous expression and is distinct from Tbx18 and Wt1b in epicardium. The authors can just use Tcf21 as an epicardial marker.

A: Thank you for this suggestion. We have deleted “pan-” and revised the text to use *tcf21* as an epicardial marker.

2. The authors used Pdgfrb as a well-established mural cell marker. They further described mural cells as Tcf21+Pdgfrb+ cells. This is quite confusing. Does this mean that TCF21 is a mural cell marker? In fact, Tcf21 is also a well known fibroblast marker (Acharya et al. 2012, Kanisicak 2016). Perdurance of nuclear fluorescent protein (tcf21:H2R) could be a reason why most mural cells around the vessels show a lower level of Tcf21 reporter signal where most likely Tcf21 is not expressed.

Although scRNAseq data from sorted Tcf21:nucEGP cells, (Fig. 2H) mural cell cluster shows moderate Tcf21 expression along with different mural cell and smooth muscle marker expressions (Fig. S2C). One noticeable information here is that Tcf21 expresses almost in all cell types. The Tcf21 UMI/mRNA level in the mural cell cluster (cluster 6) and in the cardiomyocyte cluster (cluster 10) seems comparable (Fig. S2A, S2E). Also, in differentially expressed gene list in cluster 6 (mural cells), Tcf21 has negative logFold change.

The author should put less emphasis on Tcf21 expression in mural cells. The Tcf21 expression in the mural cells could be a spillage from the epicardial cells (the dominant FACS sorted population) as it happened to all other cell types (e.g. cardiomyocytes) sorted with it.

A: Thank you for the insightful comment and suggestion. We agree that the *tcf21* expression in mural cells could be caused by the perdurance of the fluorescent protein and/or spillage from the epicardial cells. We have now removed the *pdgfrb* and *tcf21:nucEGFP* expression data of uninjured heart (original Figure 1D). The original Figure 1E showing colocalizations of *pdgfrb:EGFP* and *tcf21:H2R* in the injured site has been moved to Figure 5F. In the revised manuscript, we use this colocalization result to suggest that mural cells in the wound may be originated from *tcf21*⁺ epicardial cells, which serves as a rationale for the fate mapping experiment in Figure 5G. We have now revised the text to only use *pdgfrb* as the mural cell marker throughout the manuscript. We also use “*pdgfrb:EGFP*⁺*tcf21:H2R*⁺” instead of “*pdgfrb*⁺*tcf21*⁺” to denote these cells shown in Figure 5F. The following sentences are included on Page 10:

*“A recent study found that *pdgfrb*⁺ cardiac mural cells are originated from the epicardium during heart development²⁹. However, whether epicardial cells give rise to *pdgfrb*⁺ cells during heart regeneration is still unclear. Gene expression analysis indicates that the branch “b” mural cells express *fn1a*, while the branch “d” cells are mostly negative (Figure 5E). Because *fn1a* expression is restricted to the injury site after 1 dpa⁴¹, we hypothesized that the branch “b” (*fn1a*⁺) mural cells in the wound are derived from aEPCs. We next crossed the *tcf21:H2A-mCherry* (or *tcf21:H2R* for short) line with a *pdgfrb:EGFP* reporter²⁹. Upon heart injury, we observed *pdgfrb:EGFP*⁺*tcf21:H2R*⁺ cells in the wound at 7 dpa (Figure 5F), further suggesting an epicardial origin of these mural cells in the wound.”*

Besides, the cardiomyocyte cluster 10 likely represents doublets of cardiomyocytes and epicardial cells, as we noted on Page 5:

“Three clusters, including 8, 10, and 11, appear to represent contaminating non-epicardial cells (or doublets).”

3. Reduced *Tcf21:nucEGFP* signal in *Tcf21*+*Ptx3a*⁺ cells (Fig.4C) and in scRNAseq data in (Fig. 2H, 3B) may not be sufficient to establish “dedifferentiated” state. It can also mean these cells are becoming some other cell type (e.g., fibroblasts, according to recent *Col12a1a* publication, Bo Hu et al., Nature Genetics, 2022) or just the new progeny cells from epicardial cell proliferation/expansion in the injured area. The finding by Bo et al. took away some novelty from this manuscript although the authors claim that *Ptx3a*+*Col12a1b*⁺ cellular population as multipotent progenitor cells. However, it may be essential to establish aEPC is a progenitor population by other markers.

A: Thank you for this critique. We have now deleted the speculation of “dedifferentiation” and instead suggested this reduction in *tcf21* expression as a sign of change in cell state in line 208 on Page 9: *“This suggests changes in cell state including cell proliferation.”*

As suggested, we selected additional aEPC markers and confirmed their expression patterns. We looked for aEPC-enriched and highly specific makers. As you can see in new Figure S5, expressions of *marcksb*, *atp5mc1*, *hmgb2b*, *hsp90b1*, *loxa*, *psmb1*, and *serpinh1a* are induced by heart injury and have colocalizations with *col12a1b*^{EGFP} signals in the injury site. Thus, these genes, together with *ptx3a* and *col12a1b*, are markers of aEPCs in adult zebrafish.

4. To validate the pseudotime analysis and demonstrate aEPC differentiation into epithelial epicardial cells and mesenchymal (inner) epicardial cells, they showed Ptx3a lineage traced cellular position (epithelial vs inner) and Hapln1a expression. While cellular position and Hapln1a expression clearly demonstrate Ptx3a⁺ cells differentiation into epicardial mesenchymal cells they also should do a Podxl HCR staining to show differentiation into epithelial epicardial cells.

A: Thank you for this suggestion. We have now included HCR staining of *podxl* in the lineage tracing experiment. As you can see from the revised Figure 6G, *ptx3a*⁺ cells give rise to *podxl*⁺ epithelial epicardial cells.

5. In Fig. 4B, the authors showed at 1dpa (and also 2dpa, Fig S6B), the Tcf21⁺ epithelial epicardial layer of the entire ventricle express Ptx3a. For cell ablation they started respective treatment from 2 dpa. This makes the interpretation a bit confusing. The authors characterized molecular signature of aEPC population from the cluster 5 cells which is specific for 3dpa sample. At 3dpa Ptx3a⁺ cells are confined surrounding the injured area (Fig. 4C). 3 dpa Ptx3a⁺ population could be more specific than pan-epicardial injury responsive Ptx3a⁺ population at day 1 and 2 which are from epithelial epicardial cells. Thus broader epithelial epicardial population (beyond aEPC = cluster 5 cells) could be ablated by 2-4 dpa Mtz treatment (and would show stronger effect by more cell ablation). To follow scRNAseq data, ideally it should be 3-5 dpa MtZ treatment.

A: Thank you for this critique. Our initial rationale for starting the Mtz treatment on 2 dpa was to reach an effective *in vivo* dosage of Mtz by 3 dpa. Importantly, the mScarlet-p2a-NTR expression of the knock-in allele is extremely low at 1 and 2 dpa and is visible only after an anti-DsRed antibody staining (noted in the Figure legends). Thus, we do not expect an epicardial cell ablation on 2 dpa. Nonetheless, to further support our results, we have now switched to 3-5 dpa Mtz treatment and repeated the analyses accordingly. We also included 2 control samples: *ptx3^{RNTR};tcf21:nucEGFP* fish with vehicle treatment and *tcf21:nucEGFP* sibling fish with 5 mM Mtz treatment. As you can see in the new Figure 7, a 3-5 dpa treatment led to blocked heart regeneration, while both control groups showed comparable and successful regeneration.

Minor correction:

1. In Fig. 7 legend the model should be I (mentioned as H).

A: The model has been moved to Figure 9K. Thank you.

2. In the Fig. 2, 4, 7 the panel orders are a bit random and thus difficult to follow.

A: We have reorganized the figure panels. Thank you.

Reviewer #2 (Remarks to the Author):

This well conducted and well written study identified a transient progenitor cell population (pdgfa+halpln1a⁺ mesenchymal epicardial cells) which drives zebrafish heart regeneration. Unfortunately, very recently a paper was published by the Poss group demonstrating similar results: hapln1 defines an epicardial cell subpopulation required for cardiomyocyte expansion

during heart morphogenesis and regeneration. This paper is briefly mentioned at the end of the discussion, however, it considerably limits the news value of the present manuscript.

A: We thank the reviewer for the effort to review our manuscript and the insightful critiques to improve the work. We have now included an extensive discussion of this paper (Sun et al., 2022, PMID: 35652354) from the Poss and Wang groups in both the Results (Pages 12-13 and Figures S9C and S9D) and Discussion sections (Page 17). We want to stress that the epicardial progenitors (*ptx3a⁺col12a1b⁺*) identified in our study give rise to the *pdgfra⁺hapln1a⁺* mesenchymal epicardial cells. The value of our study is to define aEPCs as the primary cellular source of essential epicardial cell progenies and essential factors for regeneration. *hapln1a⁺* cells are among the downstream effectors of aEPC activation in supporting heart regeneration. We hope our work will lead to focused studies on activating epicardial progenitors to promote heart regeneration or repair. The following sentences are included on Pages 12-13:

“Sun et al. recently profiled single tcf21⁺ cells isolated from adult hearts upon CM ablation¹². They found that an hapln1a⁺ subset providing hyaluronic acid (HA) is required for heart regeneration and compact muscle development. We processed their dataset and checked the expression of aEPC markers. As shown in Figures S9C and S9D, ptx3a and col12a1b are expressed in the injured 7-day sample but are merely detectable in the uninjured one. Ptx3a is primarily expressed in their cluster 2, which was suggested by Sun et al. to give rise to the adjacent clusters, including the hapln1a-enriched clusters¹². This reanalysis and our results indicate the progenitor property of ptx3a⁺ epicardial cells in different heart injury models, and that the pro-regenerative hapln1a⁺ epicardial cells are progenies of ptx3a⁺ aEPCs.”

And Page 17:

“A recent study by Sun et al. demonstrated that the hapln1a⁺ epicardial cells mediate HA secretion and myocardial development and regeneration¹². Thus, both studies suggest that the entire epicardial population actively participated in the regeneration process with diverse cellular and paracrine contributions.”

The summary highlights aEPCs as a potential target for enhancing cardiac repair. This issue, however, is not further elaborated on in the discussion section. It also remains unclear whether the findings reported in zebra fish can be translated to mice and possibly to humans. scRNAseq on mice epicardial cells have been reported last year (Hesse et al 2021) and it would have been interesting compare the zebrafish data with the available mouse data set.

A: Thank you for this critique. We have now compared our results with the published mouse epicardium dataset (Hesse et al., 2021, PMID: 34152268) and included the results as a new section, “*Comparison with mouse epicardial cells upon myocardial infarction*” on Page 15 and new Figure S10. As detailed below, this comparison demonstrates both similarities and differences in epicardial populations between zebrafish and mice. It also implies that activating the progenitor state in mice has the potential to promote cardiac repair.

In the injured adult mouse heart, epicardial cells form a multi-cell layer in the wound through EMT (PMID: 23028582). Hesse et al. named these cells in the layer as epicardial stromal cells (EpiSC) and performed scRNA-seq of FACS-isolated EpiSC 5 days after myocardial infarction

(MI). The dataset comprises 11 clusters that are separated into 3 groups: I, II, and III. We reanalyzed the datasets and reproduced the published clustering result (new Figure S10A). As Hesse reported, cells in group I (expressing *Wt1*) are located in the outermost layer of the epicardium. Expression of group III markers (such as *Sfrp2*) are present throughout the activated epicardium, but mostly in the inner layers of the epicardium. Group II has both epithelial clusters (expressing *Wt1*) and inner layer clusters (expressing *Ifit3* and *Sfrp2*), and is enriched with extracellular matrix (ECM)-related pathways.

To compare the epicardial subpopulations between zebrafish and mice, we examined the expression of zebrafish cluster markers in the mouse dataset by focusing on markers that are characterized in our current study. As shown in new Figures S10B-F, the homologs of zebrafish epithelial epicardium markers *Podxl*, *Sema3d*, and *Aldh1a2* are enriched in mouse EpiSC group I. The homologs of zebrafish aEPC markers *Ptx3*, *Col12a1*, *Marcks*, *Lox*, *Hop90b1*, *Serpinh1*, and *Tmsb4x* are primarily expressed in mouse EpiSC group II. The zebrafish mesenchymal or mural epicardium markers *Hapln1*, *Pdgfra*, and *Pdgfrb* are enriched in mouse EpiSC group III. In addition, zebrafish *fn1a*, *psmb1*, and *atp5mc1* are markers for aEPCs but are also expressed in part of the epithelial epicardium cluster (Figures 3E and S5A). Similarly, the mouse homologs *Fn1*, *Psmb1*, and *Atp5g1* are highly expressed in EpiSC groups I and II (Figure S10B-E). Thus, mouse EpiSC groups I, II, and III are comparable to zebrafish epithelial, aEPC, and mesenchymal/mural subsets, respectively.

Within group II, *Ptx3* is expressed in all clusters but is mostly enriched in cluster 2 (Figure S10B and S10E). RNA velocity analysis in Hesse et al. (copied below) suggests trajectories from cluster 2 cells to cluster 4 cells. Thus, the *Ptx3*⁺ mouse EpiSCs likely give rise to other group II cells.

[redacted]

However, unlike in zebrafish, Hesse et al. found that mouse epithelial epicardial cells (*Wt1*⁺), including some group II cells that express *Ptx3* (cluster 8, see the images copied above), do not give rise to mesenchymal EpiSCs in the infarct. This observation has also been reported by Zhou et al., 2011 (PMID: 21505261) and Quijada et al., 2020 (PMID: 31999538). Although group

II of the mouse EpiSC does express markers of zebrafish aEPCs, the RNA velocity analysis in the Hesse et al. study (see above) does not show differentiation trajectories from group II to cells in other groups. These differentiation deficiencies may contribute to the limited regenerative capacity of the adult mouse heart, which warrant further genetic studies in mice. We further elaborated on the translational potential in the revised Discussion section on Page 17:

“In mammals, the adult epicardium shows analogous activation upon heart injury (such as re-activation of embryonic gene expression, proliferation, and secretion), but this activation is limited in term of mitogen secretion, EMT, and differentiation capacity^{6, 10, 59}. The similarities between zebrafish aEPCs and mouse epicardial-derived cells suggest that awakening the progenitor potential in the adult mammalian epicardium could promote cardiac repair after myocardial infarction. In all, our study has revealed the plasticity of adult epicardial cells and highlighted the aEPCs as a target for enhancing cardiac regeneration.”

Minor:

p.4, l.61: The argument that WT1 is also expressed in other non-epicardial cells also holds for *tcf21*, certainly in mice. Fig 1 shows the *tcf21* distribution in the epicardium; does it also label cardiac fibroblasts and/or endothelial cell in the deeper layers of the myocardium?

A: Thank you for this comment. Kazu et al. (PMID: 21653610) have documented that zebrafish *tcf21* reporters drive epicardium-specific expression throughout development and regeneration without labeling cardiomyocytes, endothelial cells, or endocardial cells. Currently, there is no definitive marker in zebrafish that could distinguish fibroblasts from epicardial cells. For instance, the currently used zebrafish fibroblast markers *col1a2*, *fn1a*, and *postnb* are also expressed by the epicardium, including the epithelial epicardium (PMIDs: 29610343, 34486669, 23988577, and this study). Because *pdgfra* is a cardiac fibroblast marker in mice (PMID: 31125253), some of the *tcf21⁺hapln1a⁺pdgfra⁺* mesenchymal cells may have a fibroblast identity in the adult zebrafish heart. This warrants further investigation; however, it does not affect the conclusions of our current story. We have included the following words to reflect this information on Page 7, lines 149-153:

“The hyaluronic acid-organizing factors hyaluronan and proteoglycan link protein 1a (hapln1a), the cardiac mesenchymal stem cell and cardiac fibroblast marker platelet-derived growth factor receptor alpha (pdgfra), as well as the myocardial mitogen neuregulin 1 (nrg1) mainly label clusters other than 2 and 5 (Figures 2H and 3A)^{12, 39, 40}.

And on Page 16, lines 447-451:

“The hapln1a⁺ mesenchymal subset is divided into multiple clusters, which indicates further heterogeneity. These subsets may have additional functions in supporting regeneration even if their relative proportion remains unchanged. Gene expression profiles suggest that some mesenchymal epicardial cells may have a fibroblast identity, which warrants further investigation.”

Fig. 2 B-E: cluster numbers are difficult to read when background color is dark; perhaps change to white.

A: Thank you for this suggestion. We have changed the size, color, and position of labels to improve legibility.

Fig. 2 H : *tcf21* also labels cluster 10 which are most likely cardiomyocytes. Could this be due to cell doublets which have not been excluded for the final analysis.

A: We agree that cluster 10 likely represents doublets of cardiomyocytes and epicardial cells. The following text is included on Page 5:

“Three clusters, including 8, 10, and 11, appear to represent contaminating non-epicardial cells (or doublets).”

Fig. 7: 5 mM metronidazole is a pretty high concentration and one wonders what the appropriate controls are, a structural derivative of Mtz without biological activity? In clinical medicine the use of metronidazole has many side effects! What was the effect of the “control” (Ctrl) on normal heart regeneration?

A: Thank you for this critique. Although the use of metronidazole has side effects, it has been successfully used in many tissue regeneration studies when appropriate controls are included (e.g., PMIDs: 31786069, 35652354, 25938716, 27149989, 26965370, and 27100776). We have now included two controls in the new batch of ablation experiments: *ptx3^{RNTR};tcf21:nucEGFP* fish with vehicle treatment and *tcf21:nucEGFP* sibling fish with 5 mM Mtz treatment. As shown in the revised Figure 7, both control groups showed comparable and successful regeneration, while the Mtz-treated *ptx3^{RNTR};tcf21:nucEGFP* group demonstrated blocked regeneration.

Reviewer #3 (Remarks to the Author)

In this manuscript the authors investigated the different cell types derived from the epicardium and how these behave and contribute to the regenerative response during zebrafish heart regeneration. To characterize the heterogeneous population of epicardial-derived cells (EPDCs), they performed scRNAseq on uninjured and injured hearts in which EPDCs are GFP labeled by *tcf21* expression and clustered the cells based on their transcriptomes. 8 Main clusters with EPDCs could be identified from the data. The authors focused on cluster 5 based on the expression of pro-regenerative genes (*ptx3*, *col12a1b*). Pseudotime analysis indicates that the cluster5 cells are derived from epithelial epicardial cell state and gives rise to a mesenchymal cell state, suggesting that these cells undergo EMT. Next they generated a transgenic *ptx3:Cre* line and concluded from this that *ptx3* expression precedes the expression of *hapln1a*, confirming the pseudotime results. Finally the authors ablate the *ptx3* expressing cells from the injured heart and observed a decrease in regenerative capacity indicating that *ptx3* expressing cells are important during zebrafish heart regeneration.

My main concern is that this manuscript lacks novelty. The here described heterogeneity of the epicardium and epicardial-derived cells during heart regeneration has been studied extensively and has been well established. In addition, several studies (including studies from this group) have been published already that describe scRNAseq data on these epicardial and epicardial-derived cells and characterize epicardial derived sub-populations (PMID: 26657776, pmid:

34486669, PMID: 35652354, PMID: 35088848). Furthermore, from these and other studies it has become very clear that one or several pro-regenerative subpopulation(s) are required for the regenerative response (PMID: 29610343). What is needed now and is missing from this study is a better understanding of the mechanisms by which these pro-regenerative cells stimulate cardiac regeneration.

A: We thank the reviewer for the effort to review our manuscript and the insightful critiques to improve the work. As noted in our introduction paragraph, we agree that the cellular heterogeneity of the epicardium and epicardial-derived cells has been established. We also agree that one or several pro-regenerative subpopulation(s) are required for heart regeneration. However, it was unknown about the multipotent cell state in the adult epicardium that supports successful heart regeneration. As detailed below and in the revised manuscript, the previously reported pro-regenerative epicardial subpopulations are either derived from aEPCs or are broader populations (than aEPCs) that include the aEPCs and their progenies in the regenerating heart. Another reported pro-regenerative *col1a2*⁺ population (PMID: 29610343) actually includes the entire epicardial cell population (Figure 2H).

As detailed in our answer to your question #4, the novelty of our study is to define aEPCs as the primary cellular source of essential epicardial cell progenies and paracrine factors for successful heart regeneration. We hope our work will lead to focused studies on activating epicardial progenitors to enhance heart regeneration or repair.

Specific points to the authors:

1. The authors injected ModRNA encoding Cre into pericardial cavity. Authors claim that this will only target the epicardial cells on the heart surface. However, the original protocol for pericardial injection by Bise and Jazwinska shows that chemicals injected in this way can also enter the underlying myocardium. Authors should substantiate their claims that they target only the surface epicardium and not any cells deeper into the tissue with convincing evidence or adjust their claims.

A: Thank you for this critique. In general, mRNA molecules are much larger than chemicals and proteins, and they are anticipated to have a much less diffusion effect when injected into the pericardial cavity. We used the *tcf21:loxP-BFP-Stop-loxP-mCherry-NTR* (*tcf21:Switch*) line to restrict labels to the *tcf21*⁺ cells. Although we could not rule out labels of deeper *tcf21*⁺ cells upon injection, we did not observe a single labeled mesenchymal cell in the apex half of the uninjured ventricle 10 days after injection. By contrast, 26.8% on average of labeled cells are in the deeper layers of injured ventricles at 7 dpa (10 days after injection). This new quantification result has been included as new Figure 1F. We have now revised our claim and stated that combined use of the *tcf21:Switch* line with the pericardial cavity injection of Cre modRNA injection limits our targets to the epithelial epicardium at least by day 10 after injection in the adults. The following text is included on Page 4:

“By injecting Cre modRNAs into fish carrying the tcf21:Switch line, we labeled the epithelial layer of the epicardium with mCherry (Figures 1D and 1E, uninjured). We did not observe a single labeled mesenchymal cell in the apex half of the uninjured ventricle 10 days after injection (Figure 1F, uninjured, 13 hearts analyzed). Although we could not rule out labels of deeper tcf21⁺

cells upon injection, combined use of the *tcf21:Switch* line with the pericardiac cavity injection of *Cre modRNA* injection specifically limits our labeling to the epithelial epicardium at least by day 10 after injection in the adults. To monitor EMT of epicardial cells, *Cre modRNAs* were injected 3 days before the amputation injury, and hearts were collected at 7 dpa to assess *mCherry* expression. We observed 26.8% on average of *mCherry*⁺ cells entering the mesenchymal layer (Figures 1E and 1F, 7 dpa), indicating an EMT process in which the epithelial epicardial cells give rise to the mesenchymal epicardial cells during heart regeneration.”

2. The scRNAseq data. The authors performed scRNAseq on a large number of cells at different conditions and identified different cell clusters. For a better comparison and understanding it would be important to compare these data with previously published scRNA seq data (PMID: 26657776, pmid: 34486669, PMID: 35652354, pmid: 35088848). The authors would need to reanalyse some of the published scRNAseq data and show for example how *ptx3* expression is distributed in these published data and discuss how their *ptx3* pro-regenerative subpopulation relates to the previously identified pro-regenerative subpopulations.

A: Thank you for the suggestion. We have now reanalyzed these datasets as detailed below:

Cao et al. (PMID: 26657776) reported a transcriptomic profile of 31 isolated *tcf21*⁺ cells from uninjured adult hearts. They found that *caveolin-1* (*cav1*), a pan-epicardial marker expressed in all 3 clusters, is required for heart regeneration. Our current study has included an equivalent uninjured sample with more cells and deeper analyses. Thus, we checked *cav1* expression in the current dataset. In agreement with the Cao et al. study, *cav1* expression is broadly observed across clusters (Figure S4). Thus, the *cav1*⁺ epicardium includes the aEPC population in the regenerating heart.

The Kapuria et al. study (PMID: 35088848) focused on epicardium-derived mural cells and performed scRNA-seq of FACS-isolated *pdgfrb:EGFP*⁺ cells from adult hearts upon heart amputation injury (7 dpa) together with the uninjured control. Because *pdgfrb:EGFP* only label a small portion of *tcf21*⁺ cells, their dataset does not represent the entire epicardial cell population. Nonetheless, we have reproduced their clusters and showed injured-induced *ptx3a* and *col12a1b* expression in a subset of their epicardial/EPDC/Fibroblast cluster (Figure S9A and B). As we have demonstrated in our current study, at least part of these *pdgfrb*⁺ mural cells in the injury site are derived from aEPCs (Figures 5F-H).

Sun et al. (PMID: 35652354) performed scRNA-seq of *tcf21*⁺ cells isolated from adult hearts upon cardiomyocyte (CM) ablation. Two samples were included: uninjured control and 7 days post-ablation. They found that an *hapln1a*⁺ subset that provides hyaluronic acid (HA) is required for heart regeneration and compact muscle development. We reprocessed Sun et al.'s dataset and checked the expression of aEPC markers. As shown in the new Figures S9C and S9D, *ptx3a* and *col12a1b* are expressed in the injured 7-day sample but are merely detectable in the uninjured one. *Ptx3a* is primarily expressed in their cluster 2, which was suggested by Sun et al. to give rise to the adjacent clusters, including the *hapln1a*-enriched cluster 1 (as shown in their RNA velocity result copied below). This reanalysis suggests that *ptx3a*⁺ epicardial cells are likely progenitors that give rise to HA-producing epicardial cells in the CM ablation model.

[redacted]

In our current study, we defined the *hapln1a⁺tcf21⁺* cells as the mesenchymal epicardial cells and further demonstrated that these cells are derived from *ptx3a⁺* aEPCs after amputation injury. Thus, these results indicate the progenitor property of *ptx3a⁺* epicardial cells in different heart injury models, and that the pro-regenerative *hapln1a⁺* epicardial cells are progenies of aEPCs. We included new data to show reduced number of *hapln1a⁺* cells in the wound after aEPC ablation (new Figures 8C and 8D). Based on the Sun et al. study, our results suggest that depletion of HA production contributes to the regeneration defect caused by aEPC ablation. Thus, *hapln1a⁺* cells and HA are among the downstream effectors of aEPC activation in regulating heart regeneration.

DeBakkers et al. (PMID: 34486669) found that deletion of an epicardial gene, *prrx1b*, blocked heart regeneration with increased fibrosis. They performed scRNA-seq of FACS isolated *tcf21⁺* cells from wild type and *prrx1b* mutant hearts at 7 days post cryoinjury. They found that *prrx1b* depletion increased the amount of a pro-fibrotic fibroblast population (their cluster 3). We processed the published dataset and reproduced their clusters. As shown in Figures S9E and

S9F, *ptx3a* is expressed in their clusters 2, 3, 5, and 9, which are accumulated together on the UMAP. The percentage of *ptx3a*-expressing cells slightly decreased in the mutant (41% in the mutant versus 47% in the wild type; Figure S9G), which might contribute to the impaired regeneration. *col12a1b* was not included in the published dataset; thus, we could not check its expression. To assess the relationship between aEPCs and *prrx1*⁺ pro-regenerative epicardial cells, we checked *prrx1a* and *prrx1b* expression in our dataset (our Figure S4). In agreement with DeBakkers et al.'s finding, *prrx1a* is expressed in both the outermost and inner layers of the epicardium (enriched in the aEPC and mesenchymal epicardium), while *prrx1b* expression is much lower and only detected in a few cells. These results suggest that the pro-regenerative *prrx1*⁺ epicardial cell population is a broader population (than aEPCs) that includes the aEPCs and their progenies. It would be interesting to further study whether and how *prrx1* regulates aEPCs in zebrafish heart regeneration.

Thus, we conclude that the published pro-regenerative epicardium-derived subpopulations are either derived from aEPCs (such as the *hapln1a*⁺ or *pdgfrb*⁺ subsets) or comprise an aEPC portion (such as the *prrx1*⁺ or *cav1*⁺ cells). These reanalyses have been included as Figure S9 and a new Results section, "*aEPCs are the primary source of pro-regenerative epicardial progenies and paracrine factors for regeneration*". These results, together with our report in the current study, indicate that aEPCs are the primary cellular driver of epicardium-mediated heart regeneration in zebrafish.

3. The trajectory reconstruction that is applied to the scRNAseq data is very difficult to follow. It is not clear why the auteurs did not using RNA velocity which is more commonly used to analyse directional changes in cell states.

A: We thank the reviewer for this comment and have annotated our representation of the Monocle 3 results to make them more legible and intuitively interpretable (revised Figures 5A and 5B). Trajectories are typically inferred as the optimal path(s) through the high-dimensional expression space, trying to capture how cells transition from one end of the expression continuum to the other. The general approach of tracing cell destiny in a reduced-dimensional space was initially established by the authors of the original Monocle package (Trapnell et al., 2014, *Nat Biotechnol.* PMID 24658644). In the words of one of the leaders of computational single-cell analyses, "[*Monocle*] has been successfully applied to capture branching cell differentiation trajectories and other dynamic processes in numerous biological contexts and is by far the most common approach for inferring transcriptional dynamics" (Kharchenko, 2021, *Nat Methods.* PMID: 34155396).

Following the benchmarking of dozens of trajectory-inferring algorithms by Saelens et al. (*Nat Biotechnol.* 2019, PMID: 30936559), we therefore applied the top-scoring tool, Slingshot (Street et al., 2018, *BMC Genomics.* PMID: 29914354), and Monocle in its most recent iteration (Monocle 3, which has turned to partition-based graph abstraction to reconstruct trajectories with higher resolution, Cao et al., 2019, *Nature.* PMID: 30787437). Despite the fact that the two algorithms use different assumptions and mathematical concepts, we saw great concordance between the results, most strikingly in the identification of the aEPC cluster as one major branching point (shown below for the core epicardial clusters).

While trajectory-inference methods cannot easily determine the direction of the changes, these *in silico* results formed the basis for our extensive tracing experiments that both confirmed the trajectory results and established the directions and relationships that we have highlighted in addition to the scRNA-seq based inferences.

4. The mechanism by which cluster5 EPDCs drive cardiac regenerations remains unaddressed. The authors suggest that these cells undergo EMT, but they have not analysed this further. The observation that *snail1* is expressed is suggestive but *snail1* expression is also expressed cluster 1 cells. Furthermore, what are the factors that induce the EMT and how does the EMT relate to heart regeneration. Is there less CM proliferation in the *ptx3* ablation experiment and if so why?.

A: Thank you for this critique to improve our study. As shown in the revised Figure 7E, aEPC ablation significantly suppresses CM proliferation (~54% reduction compared to the controls). Our data indicate that aEPCs are the primary cellular source of essential epicardial cell progenies and paracrine factors for successful heart regeneration. These pro-regenerative aEPC progenies include mural cells (expressing *pdgfrb*) and mesenchymal cells (secreting HA, Nrg1, and Vegfaa). (PMIDs: 25830562, 35088848, 35652354, and 30104362). We have shown that aEPCs also express reported pro-regenerative factors including *aldh1a2*, *fn1*, *fstl1*, *tmsb4x*, and *col12a1* (PMIDs: 21397850, 23988577, 26375005, 26094634, 35179181, and 27783651). We included new evidence that aEPC ablation led to reduced Nrg1 expression and the number of HA-producing mesenchymal epicardial cells (i.e., *hapln1a*⁺ cells) in the wound (Figure 8). Thus, aEPCs promote regeneration by serving as a cellular source and signaling hub. Nrg1, HA, and *hapln1a*⁺ cells are among the downstream effectors of aEPC activation in supporting heart regeneration.

Regarding the epicardial cell EMT, we further showed that heart injury induces *tgfb1a* expression in aEPCs (cluster 5, Figure 9A). *snail1a* expression is enriched in the mesenchymal epicardium, mural cells, as well as a portion of aEPCs (Figure 9A). HCR staining results showed co-expression of *tgfb1a* and *snail1a* in *col12a1b*⁺ aEPCs in the wound (Figure 9B). We reasoned that *tgfb1a* regulates epicardial EMT. Following a published protocol (PMID: 27783651), we treated fish with a Tgfβ pathway inhibitor, SB431542, from 2 to 7 dpa (Figure 9C). This treatment led to huge blood clots with reduced *pcf21*⁺ cell coverage and the number of *ptx3a*⁺ cells at 7 dpa (Figures 9D-H), which largely mimic the aEPC ablation phenotypes. HCR staining and

quantifications demonstrated a reduced number of *hapln1a*⁺ mesenchymal epicardial cells entering the wound (Figures 9I and 9J), suggesting defects of epicardial differentiation and EMT. Although it has been documented that Tgf β inhibition blocks heart regeneration (such as PMID 22513374), our results provide further cellular insights that Tgf β -regulated EMT and differentiation of aEPCs contribute to heart regeneration.

REVIEWERS' COMMENTS

Reviewer #1 (Remarks to the Author):

The authors have addressed my previous concerns. They also provided a substantial amount of new data and analyzed more previously published datasets. The current version is a comprehensive analyses of the epicardial progenitor cells that undergo EMT and contribute to heart regeneration.

Reviewer #3 (Remarks to the Author):

In the revised version of this manuscript the authors performed scRNAseq on epicardial cells from zebrafish heart with and without an injury. With this they identified an activated epicardial cell population that gives rise to most other epicardial-derived cells during regeneration and they show that this population is essential for proper regeneration.

The conclusions made by the authors are well supported by the experimental evidence described here. In addition the use of transgenic knock-in reporter lines is very elegant and very helpful to move this field forward. My questions are well addressed by the authors and the comparison with the previously published scRNAseq data sets is very informative and helps to see the relation with the other studies.

I only have some minor comments:

1. In Fig 3D,E. it is not clear from the description whether the sections are taken from injured or uninjured heart. Please explain.
2. The authors state that they have uploaded the scRNAseq data to the GEO database. When I accessed the data it looked like the authors have only uploaded the annotated data and not the raw fastq files (SRA files was not available to me). Uploading raw sequence data will greatly help others to reanalyse and reuse the data. This should be fixed.

Reviewer #1 (Remarks to the Author):

The authors have addressed my previous concerns. They also provided a substantial amount of new data and analyzed more previously published datasets. The current version is a comprehensive analyses of the epicardial progenitor cells that undergo EMT and contribute to heart regeneration.

A: We thank the reviewer for the effort to review our manuscript and the supportive comments.

Reviewer #3 (Remarks to the Author):

In the revised version of this manuscript the authors performed scRNAseq on epicardial cells from zebrafish heart with and without an injury. With this they identified an activated epicardial cell population that gives rise to most other epicardial-derived cells during regeneration and they show that this population is essential for proper regeneration.

The conclusions made by the authors are well supported by the experimental evidence described here. In addition the use of transgenic knock-in reporter lines is very elegant and very helpful to move this field forward. My questions are well addressed by the authors and the comparison with the previously published scRNAseq data sets is very informative and helps to see the relation with the other studies.

A: We thank the reviewer for the effort to review our manuscript and the supportive comments.

I only have some minor comments:

1. In Fig 3D,E. it is not clear from the description whether the sections are taken from injured or uninjured heart. Please explain.

A: These sections are taken from injured hearts at 3 days post-amputation (dpa). We have included the information in the Figure Legend.

2. The authors state that they have uploaded the scRNAseq data to the GEO database. When I accessed the data it looked like the authors have only uploaded the annotated data and not the raw fastq files (SRA files was not available to me). Uploading raw sequence data will greatly help others to reanalyse and reuse the data. This should be fixed.

A: The raw fastq files were uploaded and stored in SRA following the platform standard. Now the GEO record GSE202836 is publicly released, and the raw files are accessible.